# WHEN MAJORITIES PREVENT LEARNING: ELIMINATING BIAS TO IMPROVE WORST-GROUP AND OUT-OF-DISTRIBUTION GENERALIZATION

## ABSTRACT

Modern neural networks trained on large datasets achieve state-of-the-art (in-distribution) generalization performance on various tasks. However, their good generalization performance has been shown to be contributed largely to overfitting spurious biases in large datasets. This is evident by the poor generalization performance of such models on minorities and out-of-distribution data. To alleviate this issue, subsampling the majority groups has been shown to be very effective. However, it is not clear how to find the subgroups (e.g. within a class) in large real-world datasets. Besides, naively subsampling the majority groups can entirely deplete some of their smaller sub-populations and drastically harm the in-distribution performance. Here, we show that tracking gradient trajectories of examples in initial epochs allows for finding large subpopulations of data points. We leverage this observation and propose an importance sampling method that is biased towards selecting smaller subpopulations, and eliminates bias in the large subpopulations. Our experiments confirm the effectiveness of our approach in eliminating spurious biases and learning higher-quality models with superior in- and out-of-distribution performance on various datasets.

## 1 INTRODUCTION

Large datasets have enabled modern neural networks to achieve unprecedented success on various tasks. Large datasets are, however, often heavily biased towards the data-rich head of the distribution (Le Bras et al., 2020; Sagawa et al., 2020; 2019). That means, there are large groups of potentially redundant data points belonging to majority subpopulations, and smaller groups of examples representing minorities. Larger groups often contain *spurious biases*, i.e., *unintended* but strong correlations between examples (e.g. image background) and their label. In such settings, overparameterized models learn to memorize the spurious features instead of the core features for the majority, and overfit the minorities (Sagawa et al., 2020). As a result, despite their superior performance on in-distribution data, overparameterized models trained on biased datasets often have a poor worst-group and out-of-distribution generalization performance.

To improve the high worst-group error and of out-of-distribution generalization, techniques such as distributionally robust optimization (DRO), or up-weighting the minority groups are commonly used (Sagawa et al., 2019; 2020). However, such methods have been shown to be highly ineffective for overparameterized models in the presence of spurious features (Sagawa et al., 2020). When majority groups are sufficiently large and the spurious features are strong, overparameterized models choose to exploit the spurious features for the majorities and memorize the minorities, as it entails less memorization on the entire data. In this setting, upweighting minorities only exacerbates spurious correlations, and subsampling the majorities has been advocated for (Sagawa et al., 2020). But, this requires the groups to be specified beforehand, which is not available for real-world datasets. Besides, random subsampling of the majority groups can entirely deplete some of their subpopulations and drastically harm the in-distribution performance (Toneva et al., 2018; Paul et al., 2021).

In this work, we propose an effective way to find large subpopulations of examples (see Fig. 1), and subsample them to ensure inclusion of representative examples from all the subpopulations. We rely on the following recent observations. In the initial training epochs, the network learns important

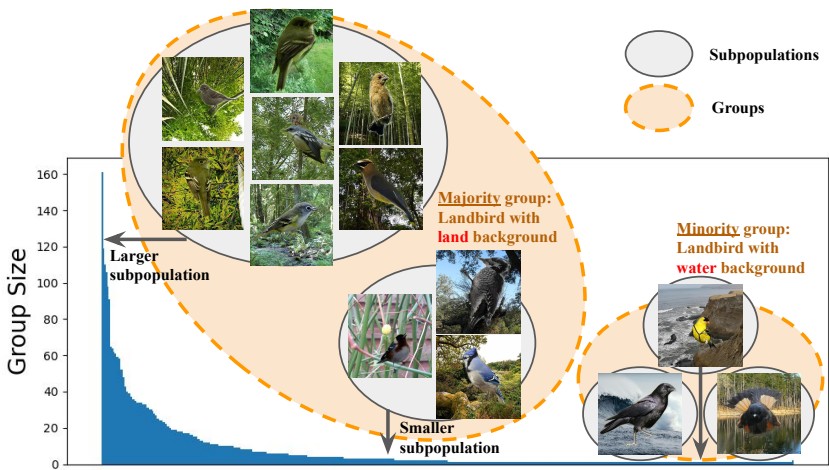

Figure 1: An illustration of large/small subpopulations within the majority/minority groups in the same class of the Waterbirds dataset (Sagawa et al., 2019).

features and the NTK undergoes rapid changes, which determine its final basin of convergence (Fort et al., 2020). This results in learning a linear function during the initial epochs, followed by learning functions of increasing complexity (Nakkiran et al., 2019). We show that large subpopulations are responsible for forming the initial linear model, by inserting large gradient forces in the first few epochs. The minorities, on the other hand, dictate the higher-complexity functions later in training. To find the large subpopulations, we track the *gradient trajectories*—the way the gradient changes— during initial training epochs. Then, we cluster similar gradient trajectories together, and employ an importance sampling method that samples data points from every cluster by a probability equal to the inverse of the size of the cluster it belongs to. This allows selecting a balanced subset from different clusters. By studying the effect of our method on the evolution of the model early during the training, we show that our method allows the model to better learn from all the subpopulations by balancing the gradient forces between different groups. This enables learning higher-quality features.

Our empirical studies confirm the effectiveness of our method in improving the worst-group and out-of-distribution generalization, while enjoying a superior in-distribution performance even when the size of the selected sample is small. Notably, on CMNIST (Alain et al., 2015) and Waterbird (Sagawa et al., 2019) datasets which contain strong spurious biases, our method achieves a comparable or even better performance than the state-of-the-art methods, which rely on the underlying group information to uniformly subsample the majority group. In addition, on CIFAR10, CIFAR100 (Krizhevsky et al., 2009), and Caltech256 (Griffin et al., 2007) our method provides a superior in-distribution performance to state-of-the-art data pruning methods, based on forgettability (Toneva et al., 2018) and El2N (Paul et al., 2021) scores, especially for small subsets. At the same time, it outperforms such methods on out-of-distribution data, CIFAR10C (Hendrycks & Dietterich, 2019).

## 2 RELATED WORK

**Data pruning for worst-group generalization.** To improve the generalization performance on minorities, preventing the model from learning spurious features is very helpful (Sagawa et al., 2019; 2020). For overparameterized models, randomly subsampling the majorities has been shown to be the most effective (Sagawa et al., 2020) than distributionally robust optimization (DRO) (Sagawa et al., 2019) and up-weighting the minority groups (Sagawa et al., 2020). However, this requires the group labels to be specified beforehand, which is not available for large real-world datasets. Besides, if the majority contains imbalanced subpopulations, random subsampling inherits similar biases. Finally, random subsampling of the majority groups can entirely deplete some of their smaller subpopulations and drastically harm the in-distribution performance, as we empirically show.

A different line of work (Sohoni et al., 2020; Nam et al., 2020; Ahmed et al., 2020; Liu et al., 2021; Creager et al., 2021; Taghanaki et al., 2021; Zhang et al., 2022; Nam et al., 2021) studies how to improve worst-group generalization without having access to group labels. These methods require training a model first to minimize the average empirical risk before training the robust model, which doubles the training time and is thus also not practical for large real-world datasets.

**Data pruning for OOD generalization.** Spurious features have been shown to also harm the out-of-distribution generalization (Le Bras et al., 2020). To alleviate this, Swayamdipta et al. (2020) proposed to train on the subset of most ambiguous instances whose true class probabilities fluctuate frequently during training, and Le Bras et al. (2020) employed AFLite (Le Bras et al., 2020) to iteratively filter highly-predictable examples by training multiple linear classifiers on different random partitions of the data. However, such methods drastically harm the in-distribution performance.

**Data pruning for in-distribution generalization.** The main idea behind all data pruning methods is to define a notion of *example difficulty*, and prune the easy-to-learn examples. Notably, Coleman et al. (2020) used a smaller trained proxy model to find the most uncertain examples to train a larger model. Toneva et al. (2018) defined a forgetting event of an example as transitioning from being classified correctly to incorrectly during training, and drop the examples with no forgetting events. Most recently, Paul et al. (2021) dropped examples with the lowest average errors (EL2N) recorded early in training and averaged over several initializations. The above heuristics require full or partial training of multiple models, can only drop a relatively small fraction of the examples, and hurt the out-of-distribution performance, as we show in our experiments. In contrast, our method can successfully alleviate bias and achieve a superior in- and out-of-distribution performance.

## 3 PROBLEM FORMULATION

Training machine learning models is often reduced to minimizing an empirical risk function (ERM). That is, the goal is to find the parameter $\boldsymbol{w}^*$ that minimizes the average error on the entire training data $\mathcal{D} = (\boldsymbol{X}, \boldsymbol{y}) = \{(\boldsymbol{x}_i, y_i)\}_{i \in V}$, where $V = \{1, \cdots, n\}$ indexes the training data. Formally,

$$\boldsymbol{w}^* = \arg\min_{\boldsymbol{w} \in \mathcal{W}} \mathcal{L}(w), \quad \mathcal{L}(\boldsymbol{w}) = \mathbb{E}_{(\boldsymbol{x}_i, y_i) \in \mathcal{D}}[l(f(\boldsymbol{w}, \boldsymbol{x}_i), y_i))], \quad (1)$$

where $\boldsymbol{w}$ is the model parameter, and $f(\boldsymbol{w}, \boldsymbol{x}_i)$ and $l(f(\boldsymbol{w}, \boldsymbol{x}_i), y_i))$ are the output of the network and the value of the loss associated to a training example $(\boldsymbol{x}_i, y_i)$, respectively. For large datasets, the average error $\mathcal{L}$ is minimized by applying (Stochastic) Gradient Descent with learning rate $\eta$ starting from a random initial point $\boldsymbol{w}_0$:

$$\boldsymbol{w}_{t+1} = \boldsymbol{w}_t - \eta \nabla \mathcal{L}(\boldsymbol{w}_t), \quad \nabla \mathcal{L}(\boldsymbol{w}_t) = \mathcal{J}(\boldsymbol{w}_t, \boldsymbol{X})^T \nabla_f l(f(\boldsymbol{w}_t, \boldsymbol{X}), \boldsymbol{y}), \quad (2)$$

where $\boldsymbol{y} = \{y_i\}_{i=1}^n$, $\boldsymbol{X} = \{\boldsymbol{x}_i\}_{i=1}^n$, and $\mathcal{J}(\boldsymbol{w}, \boldsymbol{X}) \in \mathbb{R}^{n \times m}$ is the Jacobian matrix associated with the nonlinear network $f : \mathbb{R}^d \to \mathbb{R}^o$ defined as

$$\mathcal{J}(\boldsymbol{w}, \boldsymbol{X}) = \big[ \frac{\partial f(\boldsymbol{w}, \boldsymbol{x}_1)}{\partial \boldsymbol{w}} \quad \cdots \quad \frac{\partial f(\boldsymbol{w}, \boldsymbol{x}_n)}{\partial \boldsymbol{w}} \big]^T, \quad (3)$$

and $\nabla_f l(f(\boldsymbol{w}_t, \boldsymbol{X}_S), \boldsymbol{y}_S)$ is the gradient of the loss w.r.t. the network. Furthermore, $\boldsymbol{\Theta}_t(\boldsymbol{X}, \boldsymbol{X}) = \mathcal{J}(\boldsymbol{w}_t, \boldsymbol{X}) \mathcal{J}(\boldsymbol{w}_t, \boldsymbol{X})^T$ is the empirical neural tangent kernel (NTK) (Jacot et al., 2018; Du et al., 2018), describing the evolution of the network during training by gradient descent.

**Spurious features and majority groups.** We consider a similar setting with (Sagawa et al., 2019; 2020), where each training example $(\boldsymbol{x}_i, y_i)$ is associated with a spurious attribute $a_i$ that is correlated with its label $y_i$. The examples with the same spurious attribute and label make a group $g_{j,k} \in G$, where $g_{j,k} = \{(\boldsymbol{x}_i, y_i) | i \in V, a_i = j, y_i = k\}$. The groups which contain considerably more examples than the rest are referred to as *majority groups*. For example, in the Waterbirds dataset (Sagawa et al., 2019), every example $(\boldsymbol{x}_i, y_i)$ belongs to one of the 2 classes, $y_i \in \{\text{waterbird}, \text{landbird}\}$ and the image background $a_i \in \{\text{water background}, \text{land background}\}$ is spuriously correlated with the label $y_i$. Thus, there are four groups of examples associated with every combination of spurious attribute and label, i.e., $\mathcal{G} = \{(\text{waterbird}, \text{water background}), (\text{waterbird}, \text{land background}), (\text{landbird}, \text{water background}), (\text{landbird}, \text{land background})\}$. The majority groups are (waterbird, water background), and (landbird, land background). Importantly, in this work, we assume that the groups and spurious attributes are *not known* at training time.

**Subpopulations.** Every dataset can be partitioned into $s$ different *subpopulations* of examples that are similar in terms of their effect on training, i.e. the indices of the training data $V$ can be partitioned into $V = \{V_1, \cdots, V_s\}$. For a formal definition, see Section 4.2. Note that subpopulations may represent a finer clustering compared to group clustering. Fig. 1 shows an illustration of groups vs. subpopulations for Waterbird dataset. We develop a method that automatically clusters the data and identifies *large* subpopulations.

**Objective.** Our goal is to find a subset $S \subseteq V$ of size $r = |S|$ from all training examples indexed by $V$, such that training on the subset alleviates the effect of spurious biases and improves (1) the worst-group generalization when the groups are imbalanced, or (2) out-of-distribution generalization under distribution shift. In both cases, we aim to preserve a good performance on the in-distribution data. In particular, the *worst-group* error is defined as,

$$\text{Err}_{wg} = \max_{g \in \mathcal{G}} \mathbb{E}_{\boldsymbol{x}_i, y_i | g}[y_i \neq y_f(\boldsymbol{w}, \boldsymbol{x}_i)], \tag{4}$$

where $y_f(\boldsymbol{w}, (\boldsymbol{x}_i, y_i)$ is the label predicted by the model. In other words, $\text{Err}_{wg}$ measures the highest fraction of examples that are incorrectly classified across all groups. Similarly, the *out-of distribution* (OOD) performance measures the performance of the model $f$ trained on the training set $\mathcal{D}$, and tested on $\mathcal{D}' = (c(\boldsymbol{X}), \boldsymbol{y})$, when $c$ is from a set of shifting functions $\mathcal{C}$. Formally,

$$\text{Err}_{ood} = \mathbb{E}_{(\boldsymbol{x}_i, y_i) \in \mathcal{D}'}[y_i \neq y_f(\boldsymbol{w}, \boldsymbol{x}_i)], \tag{5}$$

measures the fraction of examples that are misclassified when $(\boldsymbol{x}_i, y_i)$ is drawn i.i.d. from $\mathcal{D}'$.

## 4 Eliminating Bias in the Data

In this section, we present our main results. We start by discussing the effect of large subpopulations on early learning dynamics. Then, we explain how gradient trajectories of examples during the initial training epochs allow finding the large subpopulations. Next, we employ importance sampling to find a subset that contains a similar number of examples from subpopulations. Finally, we study how the subset found by our method affects the network's early learning dynamics.

### 4.1 Effect of Large Subpopulations on Early Learning Dynamics

Recent empirical studies on neural networks' training dynamics show that in the initial epochs the performance of a network trained by SGD can be explained by a linear classifier. Formally, if $F$ and $L$ are the corresponding random variables for the neural network and a linear model respectively, the mutual information between $F$ and $\boldsymbol{y}$ conditioned on $L$, $I(F; \boldsymbol{y} | L)$, captures the part of $F$'s success on the prediction of $\boldsymbol{y}$ in addition to $L$. Then the performance correlation between $F$ and $L$, $\mu_{\boldsymbol{y}}(F; L) := I(F; \boldsymbol{y}) - I(F; \boldsymbol{y} | L)$, is the part of $F$'s success on the prediction of $\boldsymbol{y}$ that can be explained by $L$. Nakkiran et al. (2019) show that there exists $T_0$ such that $\mu_{\boldsymbol{y}}(F_t; L) \approx I(F_t; \boldsymbol{y})$ at training step $t$ for all $t < T_0$. As training progresses, the network learns functions of increasing complexity (Nakkiran et al., 2019). Furthermore, Fort et al. (2020) show that during the first few epochs of training, neural network experience a rapid initial transient which determines the final basin of convergence. During this period, the NTK changes very rapidly and learns useful features.

First, we empirically show that large subpopulations are responsible for forming the initial linear model in the first few training epochs. Effectively, during training every example contributes to minimizing the loss by its gradient. Examples with similar gradients insert a similar force on the model and affect the model in the same direction. In the first few epochs, large gradient forces of large subpopulations highly bias the initial linear function. As the gradient forces of large subpopulations persist during training, the initially learned linear function is retained, to the point of zero training error (Nakkiran et al., 2019). As a result, large subpopulations dictate the rapid initial change of the NTK and the prominent features learned in this phase. On the other hand, smaller subpopulations have a smaller influence on the model, and require a larger number of training iterations to be learned by the higher-complexity functions that are shaped later during the training.

When the spurious bias is strong, the initial linear function is dictated mainly by the spurious feature, and persists during training. This provides a good training and generalization error on the large groups, and thus prevents learning their core features . On the other hand, on the small subpopulations, functions with much higher complexity overfit and memorize the minorities. Such functions result in a small training error but a poor worst-group generalization performance on the minorities. This further explains the observation by Sagawa et al. (2020) showing that overparameterized models memorize the spurious feature and overfit the minorities. Effectively, the spurious features in large subpopulations prevent the model from learning the core features from the data.

Fig. 2a shows that the network's prediction early in training can be well explained by a linear classifier (red). Besides, the network trained on large subpopulations can be well explained by the same

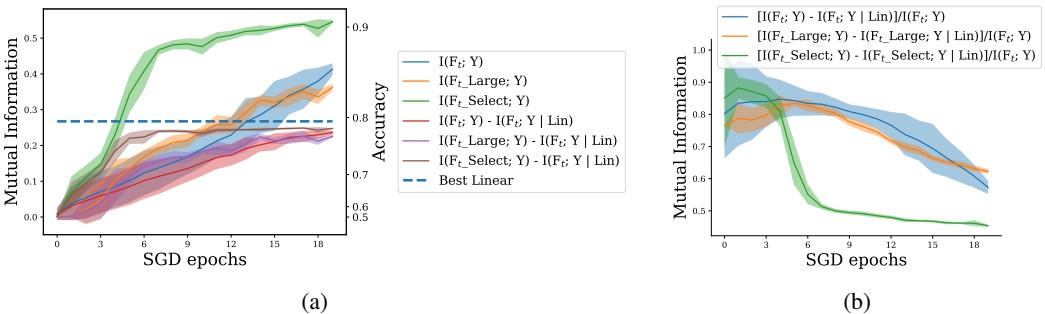

(a)                                          (b)

Figure 2: CIFAR-10/4-layer CNN. (a) mutual information between the labels, and a linear model fitted to entire training data, with networks trained on full data, large subpopulations, and our subset. (b) Fraction of network predictions explained by a linear classifier fitted to full data (blue) , large subpopulations (orange), and our subset (green). Large subpopulations shape the model initially.

linear classifier (purple). This confirms that early training dynamics are dictated by large subpopulations. We also see that the behavior of the network trained on the subset selected by our method (discussed in Section 4.3) cannot be explained by the same linear model (brown). Fig. 2b shows that the linear model explaining the network trained on full data (blue) is similar to the linear model explaining the network trained on large subpopulations (orange), but different than the one explaining the network trained on our chosen subset (green). We see that the performance of the network can be well explained by a linear mode, and the linear model fitted on large subpopulations closely matches the one fitted on the entire data. These results further confirm that the linear classifiers fitted to entire data and large subpopulations are effectively the same during the initial training epochs.

## 4.2 Finding the Large Subpopulations in Every Class

The first question we aim to answer is how to find the large subpopulations of the data, without having such labels. As discussed, larger subpopulations insert a large gradient force on the model, and are learned during the initial epochs. When an example is learned, its gradient becomes nearly zero. Hence, every example has a *gradient trajectory* interpolating between its gradient at initialization and zero. Subpopulations that affect the model similarly have a similar gradient trajectory during training. Therefore, large subpopulations with similar gradient trajectories can be identified based on their gradient trajectory during the first few epochs.

To find the large subpopulations, we cluster the gradient trajectories during the initial epochs of training. As gradients are very high-dimensional, we first reduce the gradient dimensionality to better find the clusters. To do so, we rely on the following observation: for neural networks, the variation of the gradient norms is mostly captured by the gradient of the loss w.r.t. the input to the last layer of the network (Katharopoulos & Fleuret, 2018). The above lower-dimensional gradients can be efficiently computed in a closed form, and has been used as a gradient proxy in several recent works (Mirzasoleiman et al., 2020; Paul et al., 2021; Pooladzandi et al., 2022). Formally, for every example $i$ we build its gradient trajectory by concatenating the lower-dimensional gradients during the first $t$ training epochs, i.e.,

$$\nabla_f^{0:t} l(\boldsymbol{x}_i, y_i) = [\nabla_f l(f(\boldsymbol{w}_0, \boldsymbol{x}_i), y_i), \nabla_f l(f(\boldsymbol{w}_1, \boldsymbol{x}_i), y_i), \cdots, \nabla_f l(f(\boldsymbol{w}_t, \boldsymbol{x}_i), y_i)], \quad (6)$$

where $\nabla_f l(f(\boldsymbol{w}_j, \boldsymbol{x}_i), y_i)$ is the gradient of the loss w.r.t. the input to the last layer of the network at epoch $j$ for training example $(\boldsymbol{x}_i, y_i)$. Note that as the gradient of an example depends on its label, examples from different classes do not have a similar gradient. Hence, we find similar gradient trajectories from every class separately.

Next, we cluster gradient trajectories to find the large subpopulations in every class. While any clustering algorithm can be used, we use the $k$-medoids objective to find the clusters efficiently. In particular, for $0 < \kappa < 1$, we partition a class indexed by $V_c \subseteq V$ to $k_c = \kappa \cdot |V_c|$ subpopulations, by first finding the set $S_c$ of its $k_c$ most centrally located gradient trajectories (medoids) by solving:

$$S_c^* \in \underset{S \subseteq V_c, \ |S| \leq k_c}{\arg\max} \ F(S) \quad \text{s.t.} \quad F(S) := \sum_{i \in V_c} \max_{j \in S_c} (\text{cnt} - \|\nabla_f^{0:t} l(\boldsymbol{x}_i, y_i) - \nabla_f^{0:t} l(\boldsymbol{x}_j, y_j)\|), \quad (7)$$

---

**Algorithm 1** Training without Bias

---

**Input:** Model $f$, initial epoch number $t$, subset fraction $\kappa$
**Output:** Model $f$ trained without bias
1: Train the model $f$ for $t$ epochs from $\boldsymbol{w}_0$ and save gradient trajectories $\nabla_f^{0:t} l(\boldsymbol{x}_i, y_i)$ for all $i \in V$
2: **for** every class $V_c$ **do**
3:    $S_c \leftarrow \emptyset$
4:    **for** $i = 1, 2, \cdots, \kappa \cdot |V_c|$ **do**
5:       $j \in \arg\max_{e \in V \setminus S_c} F(e|S_c)$
6:       $S_c = S_c \cup \{j\}$
7:    **for** $i \in |S_c|$ **do**
8:       $V_{c,i} = \{j \in V_c | i = \arg\min \|\nabla l_f(\boldsymbol{w}, \boldsymbol{x}_i) - \nabla l_f(\boldsymbol{w}, \boldsymbol{x}_j)\|$
9:    **for** $j \in V$ **do**
10:       $w_j = |V_{c,i}|$ s.t. $j \in V_{c,i}$
11:       $p_i = u_i^{1/w_i}$ s.t. $u_i \in (0,1)$ is a uniform random number
12:    $S = \{r$ examples with the largest $p_i\}$
13: Train the model $f$ from $\boldsymbol{w}_0$ on $S$

---

where cnt is a large constant. Then to find the subpopulations, we assign every example to the medoid $j \in S$ with the most similar trajectory. This partitions examples in class $V_c$ to $k_c$ subpopulations $V_c = \{V_{c,1}, \cdots, V_{c,k_c}\}$, where $V_{c,j} = \{i \in V_c | j = \arg\min_{s \in S_c} \|\nabla_f^{0:t} l(\boldsymbol{x}_i, y_i) - \nabla_f^{0:t} l(\boldsymbol{x}_r, y_s)\|\}$.

The maximization problem (Eq. (7)) is NP-hard. However, since the $k$-medoids objective is monotone and submodular[1], a near-optimal solution of size $k$ can be found efficiently in $\mathcal{O}(|V| \cdot k)$ time. For maximizing a monotone submodular function, the greedy algorithm provides a $(1 - 1/e)$ approximation guarantee (Wolsey, 1982). The greedy algorithm starts with the empty set $S_0 = \emptyset$, and at each iteration $l$, chooses an element $e \in V$ such that $S_l = S_{l-1} \cup \{\arg\max_{e \in V} F(e|S_{l-1})\}$.

### 4.3 BALANCING THE SUBPOPULATIONS

To alleviate the bias of the large subpopulations and enable effective learning of core features, we aim to prune the large gradient trajectory clusters formed in initial epochs. This prevents the initial linear model from being biased toward the large subpopulations. In doing so, we allow the initial linear model to capture the complexity in different subpopulations, and learn the core features instead of the spurious features of the majorities. Hence, the model obtains a better generalization performance on minorities and out-of-distribution data. However, this should be done carefully as over-pruning the large subpopulations prevents them from participating in forming the initial model. This drastically harms the in-distribution generalization performance of the model.

To address this, we employ an importance sampling method on the union of the subpopulations of all classes, to select every example by probability equal to the inverse of the size of the subpopulation it belongs to. In particular, we weigh every example $i \in V_{c,j}$ by the size of the cluster $j \in S_c$ it belongs to, i.e., $w_i = |V_{c,j}|$. Then, we use the algorithm of Efraimidis & Spirakis (2006) to select a sample with probabilities equal to $p_i = 1/w_i$, without replacement. The sampling procedure works as follows. For each example $i$ in the dataset, we independently generate a uniform random number $u_i \in (0,1)$ and calculate $q_i = u_i^{1/w_i}$. Examples that possess the $r$ largest $q_i$ form the final subset $S$.

Our sampling method biases the sample selection towards the smaller subpopulations, and drops many examples from the larger subpopulations. However, it still preserves the patterns in larger subpopulations, by including a smaller number of their examples in the sample. Effectively, our method balances the gradient forces between different subpopulations. This increases the strength of the core gradient vs. the spurious gradient. In doing so, it allows different subpopulations to participate in forming the initial linear model and dictate a more generalizable basin in which the model can be further fine-tuned. Hence, it enables better learning of the core features.

The pseudocode is illustrated in Algorithm 1.

---

[1] A set function $F : 2^V \to \mathbb{R}^+$ is *submodular* if $F(e|S) = F(S \cup \{e\}) - F(S) \geq F(T \cup \{e\}) - F(T)$, for any $S \subseteq T \subseteq V$ and $e \in V \setminus T$. $F$ is *monotone* if $F(e|S) \geq 0$ for any $e \in V \setminus S$ and $S \subseteq V$.

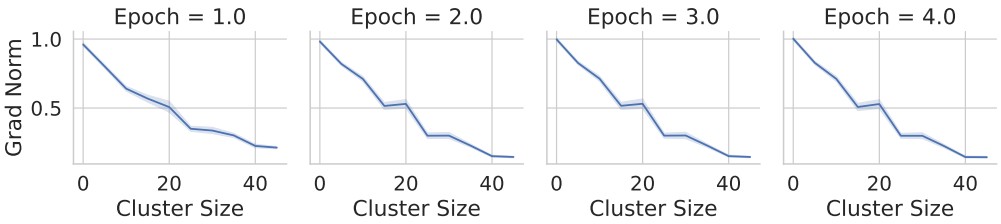

Figure 3: Average gradient norm per cluster vs. cluster size.

## 4.4 EFFECT OF PRUNING ON EARLY NETWORK EVOLUTION

Next, we take a closer look at the effect of our method on the evolution of the model. In particular, we show that training on the subset $S$ selected by our method decreases the speed of learning on large subpopulations and lets the other groups have a larger contribution to the initial phase of learning.

When the model is trained on the subset $\mathcal{D}_S = (\boldsymbol{X}_S, \boldsymbol{y}_S)$, the weight evolution over one step can be written as

$$\Delta_S \boldsymbol{w}_t = -\nabla \mathcal{L}(\boldsymbol{w}_t, \boldsymbol{X}_S) = -\eta \mathcal{J}(\boldsymbol{w}_t, \boldsymbol{X}_S)^T \nabla_f l(f(\boldsymbol{w}_t, \boldsymbol{X}_S), \boldsymbol{y}_S). \tag{8}$$

Furthermore, the network evolution can be approximated using a first-order Taylor expansion, i.e.,

$$\Delta_S f(\boldsymbol{w}_t, \boldsymbol{X}) = \mathcal{J}(\boldsymbol{w}_t, \boldsymbol{X}) \Delta_S \boldsymbol{w}_t = -\eta \mathcal{J}(\boldsymbol{w}_t, \boldsymbol{X}) \mathcal{J}(\boldsymbol{w}_t, \boldsymbol{X}_S)^T \nabla_f l(f(\boldsymbol{w}_t, \boldsymbol{X}_S), \boldsymbol{y}_S) \tag{9}$$

$$= -\eta \boldsymbol{\Theta}_t(\boldsymbol{X}, \boldsymbol{X}_S) \nabla_f l(\boldsymbol{w}_t, (\boldsymbol{X}_S, \boldsymbol{y}_S)), \tag{10}$$

where $\boldsymbol{\Theta}_t(\boldsymbol{X}, \boldsymbol{X}_S) = \mathcal{J}(\boldsymbol{w}_t, \boldsymbol{X}) \mathcal{J}(\boldsymbol{w}_t, \boldsymbol{X}_S)^T$ is the empirical neural tangent kernel, describing the evolution of the network when training only on the subset $S$. The following Lemma quantifies the effect of pruning the large subpopulation on the model evolution at one training step.

**Lemma 4.1** *Training on the subset $S$ sampled from $\zeta$ subpopulations found by our method, with learning rate $\eta \le 1/\|\mathcal{J}(\boldsymbol{w}_t, \boldsymbol{X})\|$ changes the predictions of the model at every step by at most:*

$$\|\Delta f(\boldsymbol{w}_t, \boldsymbol{X}) - \Delta_S f(\boldsymbol{w}_t, \boldsymbol{X})\| = \eta \|\boldsymbol{\Theta}_t(\boldsymbol{X}, \boldsymbol{X}) \nabla_f l(\boldsymbol{X}, \boldsymbol{w}_t) - \boldsymbol{\Theta}_t(\boldsymbol{X}, \boldsymbol{X}_S) \nabla_f l(\boldsymbol{X}, \boldsymbol{w}_t)\| \tag{11}$$

$$\le \sum_{z \in [\zeta]} |\alpha'_z - \alpha_z| \cdot \| \max_{j \in V_z} \nabla l(f(\boldsymbol{w}_t, \boldsymbol{x}_j), y_j)\|, \tag{12}$$

*where $\alpha_z = |V_z|$ is the size of subpopulation $V_z$, and $\alpha'_z = |V_z \cap S|$ is its size in the subset $S$.*

The proof can be found in Appendix A.1.

Lemma 4.1 upper-bounds how training on the subset found by our method changes the effect of different subpopulations on the model predictions. When the subpopulations are approximately balanced, we have $\alpha'_z \approx \kappa \alpha_z$. Thus, training on the subset $S$ yields similar network evolution to that of the full data, and only scales down the learning rate. However, when subpopulations are imbalanced, it effectively decreases the gradient force of large subpopulations by $|\alpha_z - \alpha'_z| \cdot \| \max_{j \in V_z} \nabla_f l(f(\boldsymbol{w}_t, \boldsymbol{x}_j), y_j)\|$. Effectively, this reduces the speed of learning and bias of such subpopulations on the model. On the other hand, our importance sampling method preserves the small subpopulations, i.e., $\alpha_z \approx \alpha'_z$ and maintains their original gradient force on the model. Therefore, our subset balances the gradient forces and let different subpopulations participate in forming the lower-complexity models in the initial epochs. As Fig. 3 shows, individual examples in large subpopulations have a smaller gradient norm. Hence, a larger number of them can be pruned without significantly affecting the model. However, entirely dropping the large subpopulations have a larger cumulative effect compared to dropping a smaller number of examples in small subpopulations with larger norms. Hence, it drastically harms the in-distribution performance.

## 5 EXPERIMENTS

In this section, we evaluate the effectiveness of our method in assisting neural networks to learn better features. In particular, we consider the following two scenarios. First, we apply our method to improve the worst-group generalization performance, when training data contains spurious correlation. Then, we consider the application of our method to improve out-of-distribution performance, under distribution shift. In both cases, we also compare the in-distribution generalization performance of the networks trained on our subset vs full training data.

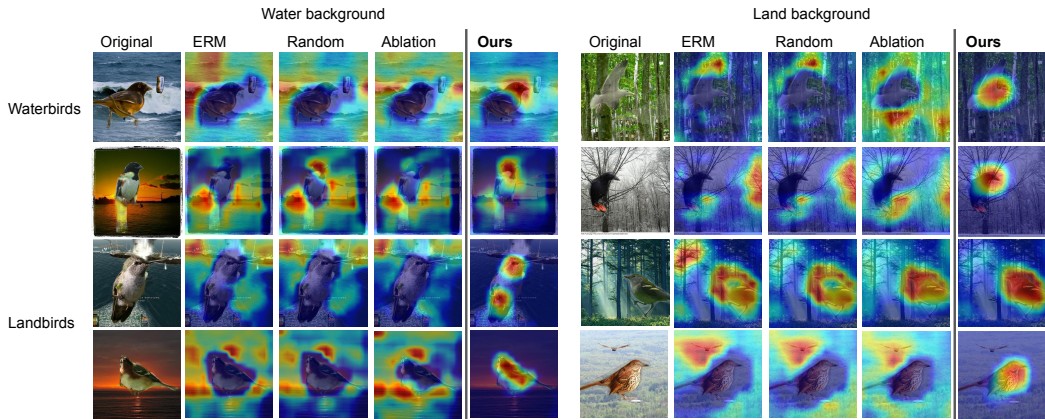

Figure 4: GradCAM Visualization. Warmer colors denote higher saliency, and correspond to the pixels that are more important in making the final classification. The subset found by our method allows the model to effectively learn the core features instead of spurious ones.

Table 1: Worst-group and average accuracy (%) on Waterbird and CMNIST 20% subsets.

| | Group Labels | Train Twice | CMNIST | | Waterbirds | |
|---|---|---|---|---|---|---|
| | | | Worst-group | Average | Worst-group | Average |
| ERM | No | No | 56.4±18.8 | 89.8±1.4 | 67.4 ± 2.9 | **97.7 ± 0.1** |
| Balanced | Yes | No | **92.5±1.0** | **95.1±1.0** | **89.1 ± 0.4** | 90.7 ± 0.9 |
| Upweight | Yes | No | 0.0±0.0 | 21.2±0.0 | 78.0 ± 2.0 | 86.8 ± 1.8 |
| Random | No | No | 56.7±3.2 | 90.0±0.3 | 67.7 ± 2.0 | **97.7 ± 0.1** |
| Ablation | No | No | 0.0±0.0 | 20.0±0.1 | 20.0 ± 1.3 | 93.7 ± 0.2 |
| **Ours-Prune** | No | No | 85.5±4.9 | 94.2±0.6 | 77.2 ± 1.3 | **97.7 ± 0.2** |
| **Ours-Reweight** | No | No | 91.6±1.0 | 96.0±0.7 | 79.1 ± 0.9 | 97.4 ± 0.1 |

## 5.1 Worst-group Generalization in Presence of Spurious Correlation

First, we evaluate the worst-group generalization performance of a model trained on our subset vs full data, in presence of spurious correlation. We record gradient trajectories during the initial 4 epochs and select 10% training examples as the subset. The reported results are averaged over 3 runs.

**Datasets & Models.** We apply our method to the Colored-MNIST and Waterbirds datasets. The Colored-MNIST dataset is a synthetic dataset derived from MNIST (LeCun et al., 1998). It was first proposed in (Alain et al., 2015) as a binary classification task that contains spurious correlations— the grey-scale digits are changed to colors that are strongly correlated with the labels. We use a 5-layer CNN with 2 convolutional layers and 3 fully-connected layers. The Waterbirds dataset is introduced by Sagawa et al. (2019) to study the spurious correlation between the background and the foreground in image recognition. Species in Caltech-UCSD Birds-200-2011 (CUB-200-2011) dataset (Wah et al., 2011) are grouped into two classes, waterbirds and landbirds. All birds are then cut and pasted onto new background images, with waterbirds more likely to appear on water and landbirds having a higher probability on land. There are 4795 training examples in total, 3498 for landbirds with land background, 184 for landbirds with water background, 56 for waterbirds with land background, and 1057 for waterbirds with water background. We use a pretrained ResNet-50 model.

**Baselines.** Empirical risk minimization (**ERM**) trains on all data, **Random** selects a subset uniformly at random, **Upweight** weights every example by the inverse of the group size, **Balanced** samples an equal number of examples from different groups. **Ablation.** To show the failure mode of random sampling when the majority has imbalanced subpopulations, we modify the dataset to make it more imbalanced, by pruning smaller clusters.

**Evaluation metrics.** We use two metrics proposed in Sagawa et al. (2019), namely worst-group accuracy and adjusted average accuracy. Worst-group accuracy is the minimum accuracy across all groups, and Adjusted average accuracy is the average accuracy over groups weighted by their size.

**Results.** Table 1 shows that the models trained on subsets found by our method obtain the highest worst-group and in-distribution test accuracy, when compared with baselines that do not require group labels. Besides, our method achieves a comparable performance to those that use the group information, and even outperforms them on the Waterbird dataset. We note that having group labels is not available in real-world datasets. Methods that do not rely on group labels, including our method, do not require knowing the minority groups. Therefore, they are more practical in realistic settings.

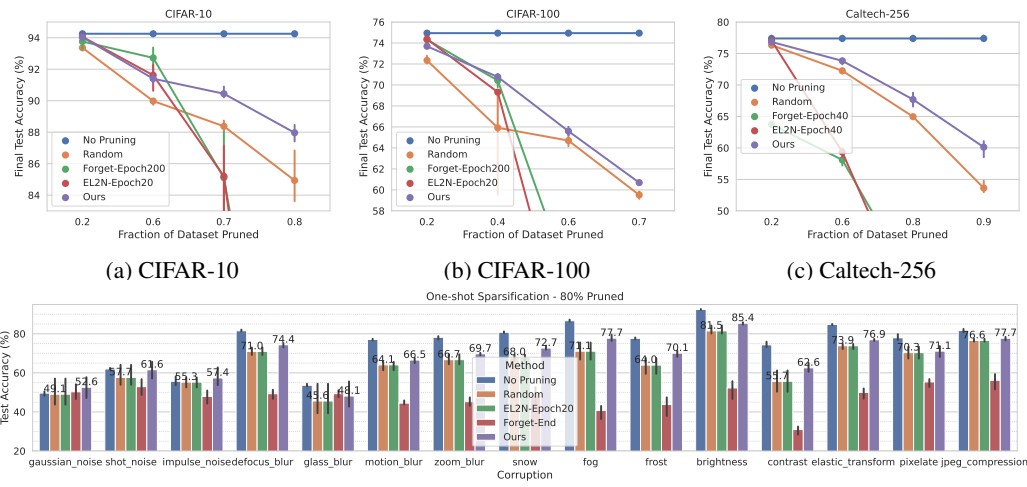

(a) CIFAR-10  (b) CIFAR-100  (c) Caltech-256

(d) Out-of-distribution performance on CIFAR-10C

Figure 5: In-distribution and out-of-distribution generalization. (a)-(c) final test accuracy on a standard (in-distribution) test set, (d) out-of-distribution performance for subset of size 20%.

**GradCam.** Fig. 4 demonstrates GradCAM (Selvaraju et al., 2017) visualizations depicting saliency maps for samples from the Waterbirds dataset with water and land backgrounds. Warmer colors denote higher saliency, suggesting that the model considered these pixels more important in making the final classification measured by gradient activations. We see that the subset found by our method allows the model to learn the core features much better than ERM and Random baselines.

## 5.2 OUT-OF-DISTRIBUTION GENERALIZATION UNDER DISTRIBUTION SHIFT

Next, we empirically evaluate the in-distribution. and out-of-distribution performance of our method under distribution shift. The results are based on 3 independent runs, each with a different mini-batch order and initial parameter values. We record gradient trajectories during the initial 4 epochs.

**Datasets.** We apply our method to CIFAR-10 and CIFAR-100 (Krizhevsky et al., 2009), and Caltech-256 (Griffin et al., 2007). In particular, we keep the number of training iterations fixed (78k for CIFAR-10 and CIFAR-100, and 4.8k for Caltech-256) as we vary the size of the subset.

**Baselines.** We compare our method with Random sampling, and the state-of-the-art baselines for in-distribution data pruning, based on EL2N (Paul et al., 2021), forgetting scores (Toneva et al., 2018). The EL2N score of a training example $i$ is defined as $\mathbb{E}\|\nabla_f l(\boldsymbol{w}, \boldsymbol{x}_i) - y_i\|_2$. We calculate EL2N after 20 epochs of training and average it over 10 different runs, as this is shown to be the most accurate. Forgetting score of an example is the number of times the examples are misclassified after being correctly classified during the entire training. We calculate the number of forgetting events for each training example by averaging over 5 runs of 200 epochs, as suggested by Toneva et al. (2018).

**Results.** Fig. 5 (a), (b), (c) show that on different datasets, training on the subset selected by our method gives much higher in-distribution test accuracy than Random, and EL2N or forgetting scores particularly when the subset is small. Note that El2N and forgettability baselines use more information over many training epochs and multiple runs. Importantly, Fig. 5 (d) confirms that our method outperforms the baselines on CIFAR-10C (Hendrycks & Dietterich, 2019), with distribution shift. We train on our downsampled CIFAR-10 training set, and test on CIFAR-10-C (Hendrycks & Dietterich, 2019), a collection of OOD test sets for CIFAR-10. For each corruption type, we report the average test accuracy over 5 different intensity levels. Our method can achieve at least 2% higher test accuracy than other baselines. For some corruption types (Gaussian noise, shot noise, and impulse noise), our performance is even on par with or surpasses training on the full data.

## 6 CONCLUSION

We showed that larger subpopulations containing spurious biases prevent learning high-quality features. We showed that large subpopulations can be identified by tracking gradient trajectory of examples in initial epochs. Then, we proposed an importance sampling method to balance the subpopulations and ensure inclusion of representative examples from all the subpopulations. Our experiments confirmed the effectiveness of our approach in eliminating spurious biases and learning higher-quality models with superior in- and out-of-distribution performance on various datasets.

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

# A  APPENDIX

## A.1  PROOF OF LEMMA 4.1

The logits evolution at one step can be written as:

$$\|\Delta f(\boldsymbol{w}_t, \boldsymbol{X}) - \Delta_S f(\boldsymbol{w}_t, \boldsymbol{X})\| = \eta \|\boldsymbol{\Theta}_t(\boldsymbol{X}, \boldsymbol{X}) \nabla_f l(\boldsymbol{X}, \boldsymbol{w}_t) - \boldsymbol{\Theta}_t(\boldsymbol{X}, \boldsymbol{X}_S) \nabla_f l(\boldsymbol{X}, \boldsymbol{w}_t)\| \tag{13}$$

$$= \eta \|\mathcal{J}(\boldsymbol{w}_t, \boldsymbol{X}) \mathcal{J}(\boldsymbol{w}_t, \boldsymbol{X})^T \nabla_f l(\boldsymbol{X}, \boldsymbol{w}_t) - \mathcal{J}(\boldsymbol{w}_t, \boldsymbol{X}) \mathcal{J}(\boldsymbol{w}_t, \boldsymbol{X}_S)^T \nabla_f l(\boldsymbol{X}_S, \boldsymbol{w}_t)\| \tag{14}$$

$$\leq \eta \|\mathcal{J}(\boldsymbol{w}_t, \boldsymbol{X})\| \cdot \|\mathcal{J}(\boldsymbol{w}_t, \boldsymbol{X})^T \nabla_f l(\boldsymbol{X}, \boldsymbol{w}_t) - \mathcal{J}(\boldsymbol{w}_t, \boldsymbol{X}_S)^T \nabla_f l(\boldsymbol{X}_S, \boldsymbol{w}_t)\| \tag{15}$$

$$\leq \|\sum_{i \in V} \nabla l(f(\boldsymbol{w}_t, \boldsymbol{x}_i)) - \sum_{j \in S} \nabla l(f(\boldsymbol{w}_t, \boldsymbol{x}_j))\| \tag{16}$$

$$\leq \|\sum_{z \in [\zeta]} \sum_{j \in V_z} \nabla l(f(\boldsymbol{w}_t, \boldsymbol{x}_j), y_j)\| \tag{17}$$

$$\leq \sum_{z \in [\zeta]} \sum_{j \in V_z} \|\nabla l(f(\boldsymbol{w}_t, \boldsymbol{x}_j), y_j)\| \tag{18}$$

$$\leq \sum_{z \in [\zeta]} |\alpha'_z - \alpha_z| \cdot \|\max_{j \in V_z} \nabla l(f(\boldsymbol{w}_t, \boldsymbol{x}_j), y_j)\|, \tag{19}$$

where Eq. equation 14 holds because $\eta \leq 1/\mathcal{J}(\boldsymbol{w}_t, \boldsymbol{X})$.

## A.2  EXPERIMENTATION DETAILS

### A.2.1  DATASETS

**CMNIST**  We construct a colored MNIST dataset with spurious correlations by using colors as the spurious attributes as the following. First, we define an image classification task with 5 classes by mapping every 2 consecutive digits (0 and 1, 2 and 3, 4 and 5, 6 and 7, 8 and 9) into the same class. We use the official test split of MNIST, randomly select 50k examples from the train split as the training set, and then use the rest 10k samples in the train split as the validation set.

Then, for each class $y_i$, we color the foreground of $p_{corr,i}$ fraction of training examples with color $a_i$ from the set of colors $\mathcal{A}=\{$#ff0000, #85ff00, #00fff3, #6e00ff, #ff0018$\}$ represented by their hex codes. We call this fraction of data the majority group of class $y_i$. The higher the $p_{corr,i}$, the stronger the spurious correlation between the class $y_i$ and the spurious attribute $a_i$. For the rest $1-p_{corr,i}$ training examples, we color them with a random color from $\mathcal{A} \setminus a_i$. In Fig. 6, we visualize examples in 5 classes with 5 colors and highlight the majority groups with white bounding boxes. In our experiments, we used $p_{corr} = [0.995, 0.95, 0.9, 0.8, 0.6]$ to construct spurious correlations with different strengths and groups with different sizes.

## A.3  ADDITIONAL EXPERIMENTS

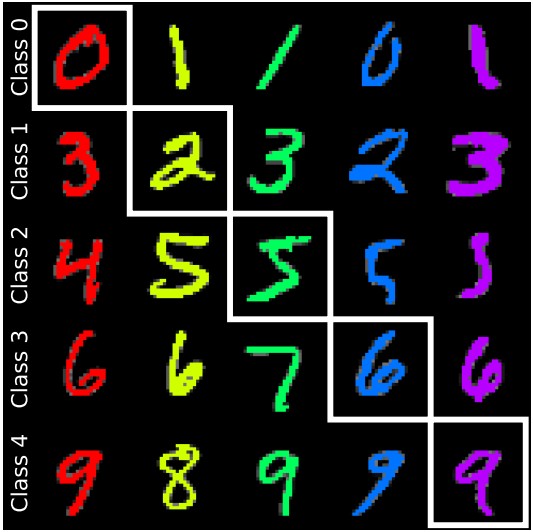

Figure 6: Visualization of the CMNIST dataset. We combine every 2 digits into one class and used 5 different colors to construct groups. This figure shows one example for each of the 25 groups. Groups with white bounding boxes are the majority groups. See Appendix A.2.1 for more details.

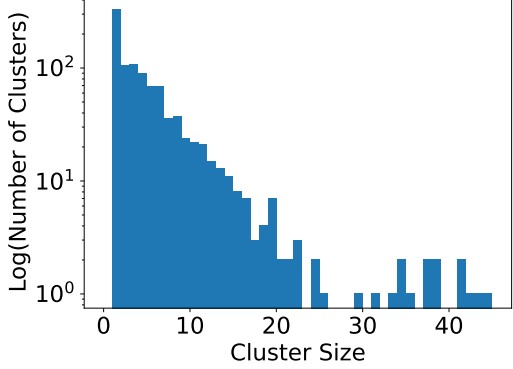

Figure 7: Distribution of cluster sizes, for Fig. 2

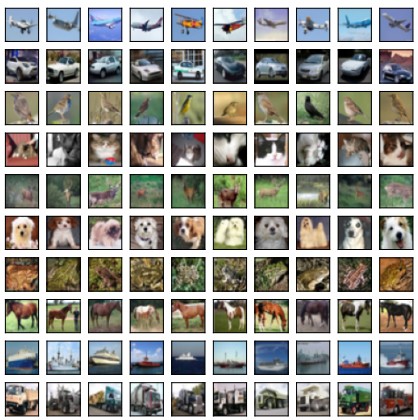

(a) CIFAR-10 images in large clusters. Examples are similar to each other.

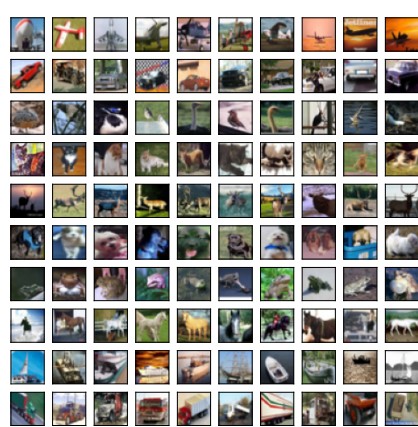

(b) CIFAR-10 images in small clusters.

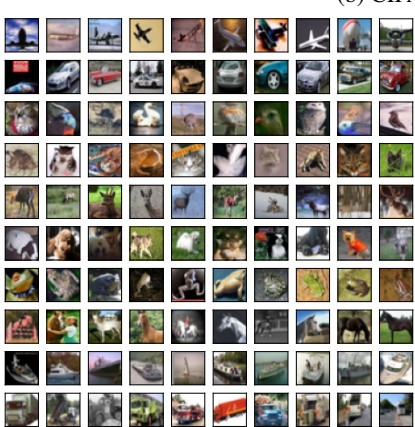

(c) CIFAR-10 images sampled by our algorithm.

Figure 8: Images in large, and small subpopulations of CIFAR-10, vs our subset used for OOD evaluation in Fig. 5d.

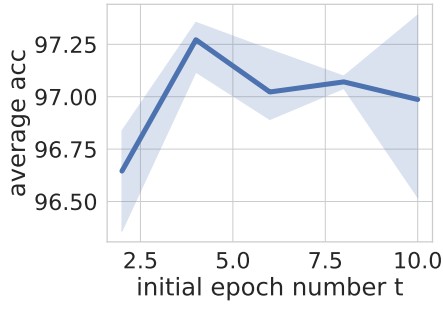

(a) Average accuracy (%) vs initial epoch number.

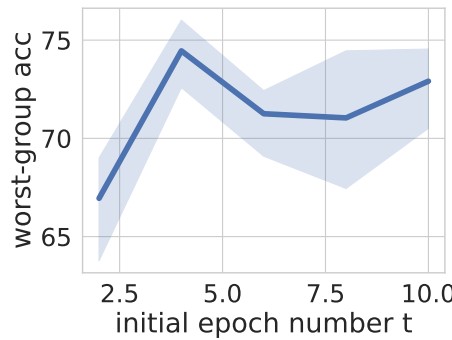

(b) Worst-group accuracy (%) vs initial epoch number.

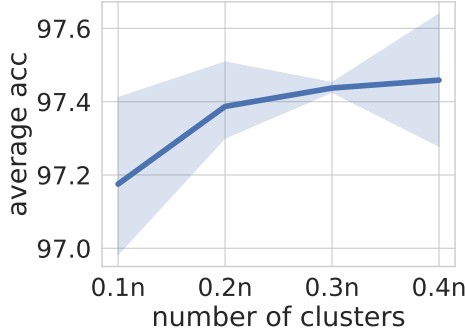

(c) Average accuracy (%) vs number of clusters.

(d) Worst-group accuracy (%) vs number of clusters.

Figure 9: Average and worst-group accuracy on Waterbirds dataset for different initial epoch number $t$ (defined in Algorithm 1) and different numbers of clusters.

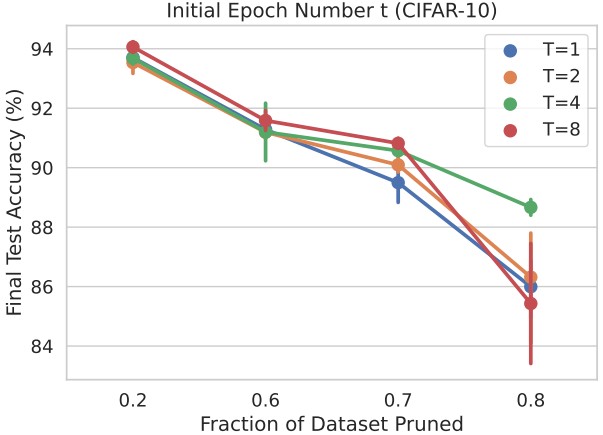

Figure 10: Test accuracy on CIFAR-10 dataset for different initial epoch number $t$ (defined in Algorithm 1).

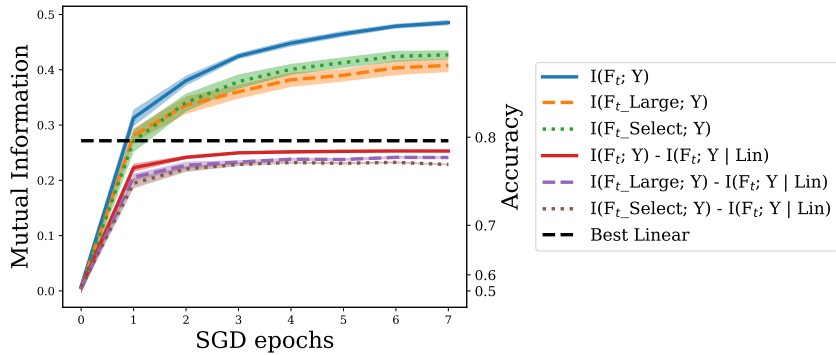

Figure 11: Mutual information between a linear model trained on full CIFAR-10 and ResNet-18 models trained on full or subsets of CIFAR-10. See the caption for Fig. 2a and Section 4.1 for more details.

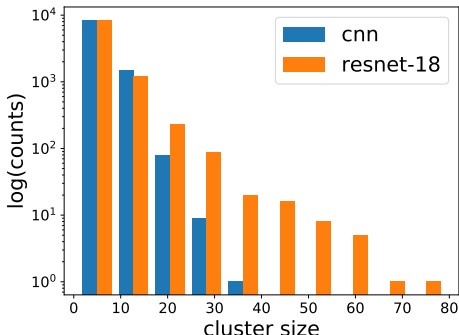

Figure 12: Comparison of gradient trajectory clusters for a 4-layer CNN vs ResNet-18.

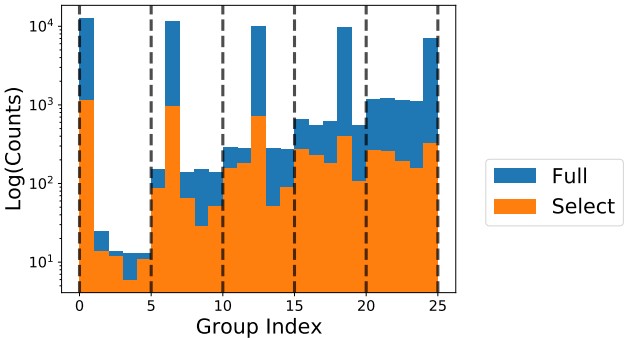

Figure 13: Number of examples in different groups of CMNIST and examples subsampled with our algorithm. Each bar represent one group and the dashed lines separate the 5 classes. Our algorithm effectively pruned the large groups.

