# OpenReview forum: "When Majorities Prevent Learning: Eliminating Bias to Improve Worst-group and Out-of-distribution Generalization"
_ICLR.cc/2023/Conference — Submitted to ICLR 2023_

### Official Review · Reviewer_BcrE · 2022-10-13

**Confidence:** 4
**Correctness:** 2
**Technical Novelty And Significance:** 2
**Empirical Novelty And Significance:** 2
**Recommendation:** 3

**Clarity, Quality, Novelty And Reproducibility:**

### Clarity
Bad.

The descriptions in Section 4.1 (motivation part) are messy and unclear, sometimes even not readable (see comments later in "Writing"). First, _mutual information_ is never formally defined or described in the text, but appears in Figure 2 -- how do you compute (or estimate) the mutual information? Is the computation the same across methods?

The overall writing quality is bad. The writing needs to be significantly improved.

Please refer to the weaknesses part for a complete review.


### Quality
Not good for the current manuscript.

There are major weaknesses in Motivation, Methods, Related work, and Exepriments.

Please refer to the weaknesses part for a complete review.


### Novelty
Fair.

I understand that the idea of leveraging gradient trajectory for subgroup discovery and robustness seems to be novel and not explored before in this field. However, the literature review is pretty limited, and it seems that the paper is not well positioned w.r.t. the literature.


### Reproducibility
Fair.

No code is provided along with the submission. The pseudo code is provided for the algorithm.

**Strength And Weaknesses:**

# Strengths
+ The idea of leveraging gradient trajectory for subgroup discovery and robustness seems to be novel and not explored before in this field.



---
# Weaknesses
Unfortunately, there are multiple major weaknesses exist in the current paper.

### Motivation
- I do not buy the motivation argument that "_large subpopulations are responsible for forming the initial linear model_".
1. Specifically, the descriptions in Section 4.1 are messy and unclear, sometimes even not readable (see comments later in "Writing"). First, _mutual information_ is never formally defined or described in the text, but appears in Figure 2 -- how do you compute (or estimate) the mutual information? Is the computation the same across methods?
2. How do you obtain "large subpopulation" in this toy example? What is "large subpopulation" and how is it defined w.r.t. CIFAR dataset?
3. Does this observation persist across different datasets and different network architectures? The authors used a 4-layer CNN -- again, the detailed architecture is not explained, and the rationale behind using this architecture is not justified (why not using a standard ResNet-18?). The obseravation could be easily driven by the fact that the capacity of the network is limited. That being said, different datasets and/or architectures might directly influence the observation, and more justifications are needed; otherwise it is not convincing.

- Related to the point above, Figure 2 in Section 4.1 is not understandable. There's no text describing the figure. Fig 2b is completely missing in the text, rather the authors refer to Figure 6 in the last paragraph of Section 4.1, which is a label distribution figure in Appendix.


### Method
-  For the importance sampling method, why directly use the inverse of the size of the subpopulation? Samples and subpopulations could have hierarchy and semantic similarities. Past works also show square-root inverse weighting can be empirially better than inverse weighting. Please explain the rationale, either theoretically or empirically, that this leads to better outcomes than other heuristic approaches (e.g., square-root inverse weighting).


### Related work
- The related work is somehow misleading and not comprehensive. The authors discuss only "Data pruning" oriented methods, either for in-distribution (avg / worst) or out-of-distribution generalization. However, the real theme of the paper is how to **improve sub-group robustness without training group information**, and this very related line of works is rather totally missing. To name a few methods along this direction (improving worst-group accuracy without group annotations): [1]-[5].

- (also weaknesses in experiments) As stated above, the line of work on worst-group accuracy **without** group annotations should also be compared in the experiments to show what advantage this paper brings, and to properly position this paper in the literature.


### Experiments
- As detailed in the "Related work" part, a major drawback of the paper is that it fails to compare with the actual line of works that is mostly related, i.e., improving worst-group accuracy without group annotations [1]-[5]. Without comparing to strong baselines as [1]-[5], the performance is not justifiable or convincing.
- Moreover, for "out-of-distribution generalization" part, the paper also failed to compare with SOTA domain generlization (DG) approaches, including [6, 7] as well as those competitive methods in the DomainBed benchmarks [8]. Again, the selected baselines are no longer SOTA, and without comparing to the aforementioned strong baselines, the results are not convincing.
- Related to the questions above, I wonder why not the authors directly test on the DomainBed benchmark. Directly testing on DG benchmarks [7] would be the most reasonable choice to fairly evaluate the proposed method to other algorithms in DomainBed.
- The choice of **first several epochs** for gradient trajectories seems to be an important hyper-parameter for the method. However, the number in the experiments seems to be rather randomly chosen; there are also no ablation studies on the number of epochs, or how does the number affect the performance.
- Related to the questions above, there are no ablation studies on the "inverse importance sampling" part. Whether this is necessary, or it is better than other heuristic approaches (e.g., square-root inverse weighting), is completely missing.
- The "gradient trajectory" idea is plausible, but also introduces potentially high computational costs. For example, ERM does not need any tracking or storing intermediate variables. Although the authors claim in the paper that the method can be implemented efficiently, I would like to see the actual computational cost w.r.t. time and memory costs. Wall-clock training time comparison as well as memory consumption to other baseline methods would be good.
- Despite of mentioning "large subpopulation" in the methods, the results never show what is the actual "large subpopulation" the model discovers, and how meaningful it is.


### Writing
- The overall writing quality is bad. The writing needs to be significantly improved.
- For example, last paragraph in page 4 is not even self-contained or readable. The first sentence in this paragraph is not grammarly correct. Moreover, "Fig. 6" mentioned in the text refers to a figure in the Appendix showing the label distribution, but has nothing to do with the texts here.
- typo: line 4 page 4 - $y_f(w,x_i)$

## References
[1] Liu et al. Just Train Twice: Improving Group Robustness without Training Group Information. ICML 2021.

[2] Zhang et al. Correct-N-Contrast: A Contrastive Approach for Improving Robustness to Spurious Correlations. ICML 2022.

[3] Nam et al. Spread spurious attribute: Improving worst-group accuracy with spurious attribute estimation. ICLR 2022.

[4] Lahoti et al. Fairness without Demographics through Adversarially Reweighted Learning. NeurIPS 2020.

[5] Creager et al. Environment Inference for Invariant Learning. ICML 2021.

[6] Cha et al. SWAD: Domain Generalization by Seeking Flat Minima. 2021.

[7] Arpit et al. Ensemble of averages: Improving model selection and boosting performance in domain generalization. 2021.

[8] Gulrajani et al. In search of lost domain generalization. 2021.

**Summary Of The Paper:**

This paper studies the problem of (in-distribution) generalization when facing potential subgroups / subpopulations in the training data, without knowing the group annotations. The paper claims that tracking gradient trajectories of examples in initial epochs allows for finding large subpopulations of data points. It also proposes an importance sampling method that is biased towards selecting smaller subpopulations, which is claimed to eliminate bias in the large subpopulations. Experiments on several datasets show that the method might be effective over several baselines in both in-distribution or out-of-distribution generalization.

**Summary Of The Review:**

The paper studies an interesting yet important problem, model generalization when facing potential subgroups / subpopulations in the training data without knowing the group annotations.

The overall idea of leveraging gradient trajectory for subgroup discovery and robustness seems to be novel and not explored before in this field, which is plausible.

However, currently there are many drawbacks and weaknesses, in terms of Motivation, Methods, Related work, and Exepriments. The writing quality  is bad and needs to be significantly improved.

In summary, I recommend rejection.

---

> ### Author Response · Authors · 2022-11-19
> **Response to Reviewer BcrE**
>
> We thank the reviewer for acknowledging the novelty of our algorithm. We will address the concerns listed in the weaknesses by sections.
>
> **Motivation**
>
> Nakkaran et al, showed that during the first few epochs of training, the predictions of the model can be well explained by a linear model (this is not the claim of our work and they theoretically justified it). We argue that larger gradient clusters insert a large gradient force on the model, and hence are learned within the first few training epochs. Then, from the above two observations, we see that the large gradient clusters are responsible for forming the linear function learned during the first few epochs. We apologize if Sec 4.3 was not clear and clarify below:
> 1. For Fig. 2, we followed the same setup as Nakkaran et al. (see their Fig 4). For mutual information, we used its standard definition. Formally, for three random variables $X, Y, Z$, mutual information is defined as $I(X;Y) = H(Y)−H(Y|X)$ and the conditional mutual information between $X$ and $Y$ conditioned on $Z$ is defined as $I(X;Y|Z) = H(Y|Z) − H(Y|X,Z)$, where H is the (conditional) entropy. The mutual information between the network predictions and labels $I(F;Y)$ can be used as a measure of the network’s performance (the two measures are monotonically related). Then, the mutual information between $F$ and $y$ conditioned on $L$, i.e., $I(F; y|L)$, captures the part of $F$’s correct predictions of $y$ that is not captured by $L$. This is measured in the same way and by the same definition across methods. Fig 2 shows that predictions of the model in the initial training epochs are very correlated with (1) those that are obtained by only training on large gradient clusters, and (2) those from a linear model trained on the full data.
> 2. For large gradient clusters, we used Fig 6 in the Appendix, which shows the distribution of cluster sizes. We chose clusters of size greater than $30$ as large clusters, which is motivated by the gap between cluster sizes at $x=30$. Please note that in our method we do not need to know the threshold for large clusters, and we only used such a threshold in Fig 2 to show that the big clusters shape the linear function that is initially learned by the model.
> 3. Our observation and argument hold for different architectures and datasets. We note that different architectures have different training dynamics, which affect the gradient clusters and the initial linear function learned by the model. However, the large gradient clusters *of the corresponding architecture* are learned early and shape the initial linear model learned by *that architecture*. Hence, our argument is valid across architectures. The reviewer is right that models may have different capacities. However, the capacity of the model does not invalidate our argument. We followed Nakkaran et al in using a 4-layer CNN in Fig 2. We added new experiments in Appendix Figure 11 showing that the same argument holds when training ResNet-18 on Cifar10. Notably, *larger architectures with more capacity have larger gradient clusters*, which is in line with observations of Sagawa et al that overparameterization exacerbates spurious biases. This is because of their larger capacity to learn more difficult examples and include them in clusters with small gradient norms. We hope this is convincing.
>
> Fig 2b was mistakenly labeled as Fig 6. We apologize for the confusion and have fixed the number in the revised version. Both Fig 2a, b are explained in Sec 4.1.
>
> **Method**
>
> The objective of importance sampling with inverse of the cluster sizes is to make the clusters balanced. If examples in every cluster are sampled by probability of inverse of the cluster size, we get roughly the same number of elements in every cluster. We added Fig 13 to the Appendix confirming this for CMNIST. We also performed new experiments using square-root inverse weighting on Waterbirds and CMNIST, confirming the superiority of sampling by inverse weight probability.
>
> |      | CMNIST || Waterbirds ||
> | ----------- | ----------- | ----------- |----------- | ----------- |
> |      | Worst-group Acc | Average Acc | Worst-group Acc | Average Acc |
> | Sqrt Inverse | $72.2 \pm 2.7$ |$92.8 \pm 0.4$ | $73.4 \pm 1.1$ | $97.2 \pm 0.1$ |
> | Inverse | $\mathbf{85.5 \pm 4.9}$ | $\mathbf{94.2 \pm 0.6}$ | $\mathbf{77.2 \pm 1.3}$ | $\mathbf{97.7 \pm 0.2}$ |

---

> > ### Author Response · Authors · 2022-11-19
> > **Response to Reviewer BcrE (Cont.)**
> >
> > **Related work**
> >
> > - We thank the reviewer for bringing the papers to our attention, and have added the papers to our related work. However, we note that *our goal is orthogonal to these works*. We aim to **alleviate the bias** in the data, by finding the subpopulations that affect learning similarly, and making them balanced early in training. This is in contrast to predicting the underlying groups when such information exists (which is not the case in most real-world datasets), and making them balanced as is done in [3]. Note that even if the groups can be identified, the subpopulations within the groups may not be balanced, and hence finding the groups is not enough to alleviate the bias. Our goal is also different than [1,2,4,5] which identify the **misclassified examples** and modify the optimization to correct their predictions. Improving the performance on misclassified examples is indeed different than alleviating the bias of large subpopulations. Methods that aim to correct misclassifications may not be necessarily advantageous to improve OOD. Similarly, existing OOD methods do not necessarily improve the worst-group performance. In contrast, we showed that reducing the bias benefits both worst-group and OOD performance. Indeed, [1,2,4,5] can be applied to improve the accuracy of the misclassified examples in our unbiased subset.
> >
> > - As discussed, our work does not directly target to improve the worst-group accuracy without having the group annotations. However, we added comparison to JTT [1] and Correct-N-Contrast (CNC) [2]. CNC showed its superior performance to EIlL (EIIL is inferior to DRO) [5], and SSA [3] which predicts the group labels and makes them balanced in the best case can be as good as our balanced baseline in Table 1, which trains on the same number of samples from each group knowing the group labels. We showed a comparable performance to the balanced baseline in Table 1.
> >
> >   We also note that JTT and EIIL are very expensive as they require training the model once on the original data and another time on the upsampled data that is orders of magnitude larger (e.g. JTT augments the data with misclassified examples 100x). JTT is also sensitive to the choice of hyperparameters (the epoch when the misclassified examples are found, and the number of times they are upsampled). Hence, JTT is not scalable to large datasets. In contrast, our method is very efficient as we show below.
> >
> >   For CnC, we couldn’t replicate the numbers reported for Waterbirds in their paper with their official code, pretrained model, and instructions, so we used the reported numbers in this table. We didn’t have problems when running the CMNIST dataset. We will update once we confirm the numbers.
> >
> > |      | CMNIST || Waterbirds ||
> > | ----------- | ----------- | ----------- |----------- | ----------- |
> > |      | Worst-group Acc | Average Acc | Worst-group Acc | Average Acc |
> > | JTT | $61.6 \pm 5.6$ | $97.2 \pm 0.2$ | $86.0 \pm 1.4$ | $91.5 \pm 0.5$ |
> > | CnC | $84.2 \pm 2.4$ | $92.0 \pm 0.4$ | $88.5 \pm 0.3$ | $90.9 \pm 0.1$ |
> >
> > **Experiments**
> > - We added new ablation studies on the choice of the epoch that we select the clusters from Waterbirds and CIFAR10 in Appendix Figure 9-10. Our results show that our method is not very sensitive to the choice of this hyperparameter, and the **first few (not several) epochs** of training provide a good performance. In all our experiments we used epoch 4 to find the clusters without further tuning.
> > - Our method is actually very fast and has a computational complexity of O(nk) to find k clusters from a dataset of size n. Note that this does not involve any exact and expensive gradient computation, and hence is very efficient. We also note that training on the smaller unbiased subset found by our method is much faster than training on the entire data. In contrast, methods such as JTT require full training of the model to convergence and also increase the size of the data by orders of magnitude (e.g. 6x for the Waterbirds dataset). We report the wall-clock time for finding the clusters here. The cost of importance sampling is negligible.
> >
> > **Reproducibility**
> >
> > We uploaded the code used for our experiments in our revision.

---

> ### Author Response · Authors · 2022-12-06
> **Looking forward to further discussions with Reviewer BcrE**
>
> We would like to follow up to see if our response addressed your concerns and if there is anything else we can clarify before the deadline.
>
> We also set up the DomainBed benchmark per the reviewer’s suggestion. However, the **OOD methods in the DomainBed require multiple domains to train on and are not applicable to a single dataset**. In addition, [6, 7] require the network to be pretrained on another (larger) dataset to have a stable convergence. **In contrast, our method intends to alleviate bias while training from scratch on a single dataset**. In our work, we showed that alleviating bias when training from scratch on a single dataset helps improve the worst-group and OOD performance (e.g. on CIFAR10-C when training on CFIAR10). However, *none of our experiments use additional information from other datasets or a model pretrained on other datasets*. Our method is inherently different from those in DomainBed and it’s not clear how we can compare our method to those that require multiple domains to work.
>
> Please let us know if we can provide any further clarification. Looking forward to hearing back from you.

---

> ### Author Response · Authors · 2022-12-07
> **Looking forward to the reviewer response**
>
> We would like to follow up to see if our response addressed your concerns and if there is anything else we can clarify before the deadline.

---

> > ### Comment · Reviewer_BcrE · 2022-12-11
> > **Response to Authors**
> >
> > I appreciate the efforts the authors made during the discussion phase. I would like to thank the authors for providing additional experiments and clarifications.
> >
> > While some of the results indeed seem interesing, they unfortunately do not address my questions and concerns.
> >
> > First, I do not buy the rebuttal comments that
> > >  our goal is orthogonal to these works (in "Related work")
> >
> > The ultimate goal is still improving the worst-case generalization results without explicitly knowing the group information. I do not see the intrinsic differences. If you wanted to demonstrate that they are orthogonal indeed, you can simply combine these methods with your technique, and that we would expect it can lead to some improvements over yours.
> >
> >
> > Further, regarding the new results of CnC & JTT - if I understand the results correctly, it clearly shows the benefits over your method on worst-case performance, especially for the more realistic Waterbirds dataset. These results haven't yet been incorpated into the paper, which in my mind, are necessary.
> >
> > Also, your claim
> > >  In addition, [6, 7] require the network to be pretrained on another (larger) dataset to have a stable convergence
> >
> > is not true. Flat minima optimizers (e.g., SWA, or SWAD in [6] or method in [7]), do not require "network to be pretrained on another (larger) dataset". In fact, as their names suggested, they have nothing to do at all with training on larger dataset or not. I suggest the authors to take a careful literature review on these latest OOD methods.
> >
> > Finally, many of the concerns in my original review are not answered or addressed.
> >
> > Considering the above as well as taking other reviewer's comments into account, I decide to keep my original rating. I strongly encourage the authors to take a full pass of the manuscript and make extra careful revisions to incorporate the results and discussions with all reviewers.

---

> > > ### Author Response · Authors · 2022-12-12
> > > **Response to the reviewer**
> > >
> > > Thank you for your reply. Our answers are indeed based on the careful consideration of all the papers referred to by the reviewer. Here are some examples:
> > >
> > > - As discussed in our previous comment, all the OOD methods referred to by the reviewer required **multiple domains to work**. In contrast, our method is designed to alleviate bias in a **single dataset**. We are not sure what the confusion is here, but if the reviewer provides more explanations, we are happy to provide further clarification in the remaining time.
> > >
> > > - In contrast to what the reviewer mentioned, **[6, 7] indeed explicitly mention in their papers that they need a pretrained network**. We urge the reviewer to check the following pointers:
> > > In Appendix B.1 of [6], it is mentioned that: “We initialize our model by ImageNet-pretrained ResNet-50 and batch normalization statistics are frozen during training. ”
> > > [7] mentions in their main paper that they use pre-trained models (in fact, they mention a “pretrained” model around 40 times in their paper.)
> > > For [7], there is a direct reply from the authors on OpenReview (https://openreview.net/forum?id=peZSbfNnBp4) saying that their “proposal of SMA is specific to the case when we initialize the model using a pre-trained model”.
> > >
> > > - Regarding JTT and CnC, as we explained in our previous comment, the goal of JTT and CNC is to _**find examples misclassified**_ by an early stopped model. The hope is that the misclassified examples represent minority groups. Then, both methods rely on upsampling (or upsampling and mixing) such examples to improve the accuracy of the minorities. Such methods do extensive oversampling, and **increase the size of the training data by up to 100x**. Hence, in contrast to the reviewer’s statement, _they are not applicable to large real-world datasets_. Please refer to Appendix A of JTT for upsampling the misclassified examples in Waterbirds (~240 examples at epoch 60) by 100x, adding ~24k additional examples on top of the ~5k examples in the original training set. CnC introduced their contrastive sampling in Algorithm 2 and more details in Appendix E.2.4. From Algorithm 2, we see that for each training example, CnC first samples (M-1) anchors, then M positives, and then 2N negative. As listed in Appendix E.2.4, M and N used for different datasets range from 16 to 64 each, and thus 4*16=64 to 4*64=256 examples in total are sampled for each training example.
> > >
> > > - Our method, on the other hand, aims to find the majority groups classified correctly during the first few training epochs. In doing so, it **allows all the groups to have a similar contribution during the early learning of a model. This improves the generalization performance of the smaller groups**. Hence, our method can easily scale to very large real-world datasets. Note that the reported accuracy comparison between our methods and JTT and CnC are _not comparable, as JTT and CnC train on orders of magnitude larger data than our method_. By being orthogonal, we mean that JTT and CnC can be applied to fix the misclassifications of the model trained on the dataset debiased by our method, which significantly reduces the number of times minority examples need to be upsampled by such methods. We also note that, the performance of our method can be significantly improved by training on a data that is orders of magnitude larger than the original one (this will make our results comparable to JTT and CnC). While we cannot conduct new experiments within the remaining 1 day, we will add them to our revision. Please also note that in contrast to the reviewer's comment, Waterbirds is an artificially created dataset with only 1 minority and 1 majority group per class, and is not really a good representative of real-world datasets.
> > >
> > > We thank the reviewer again for providing valuable feedback, however, we strongly believe that our paper deserves a much higher score than 3, and we hope the reviewer can revise the score based on our clarifications.

---

> ### Author Response · Authors · 2022-12-13
> **Last follow up**
>
> We want to follow up with you for the last time before the deadline. We hope you can read our responses and consider revising your score. Thank you!

---

### Official Review · Reviewer_gnhj · 2022-10-20

**Confidence:** 4
**Correctness:** 3
**Technical Novelty And Significance:** 4
**Empirical Novelty And Significance:** 3
**Recommendation:** 6

**Clarity, Quality, Novelty And Reproducibility:**


Clarity:

x) The language is ok.

x) The structure and flow are ok.

x) I would suggest a more intuitive color code (or line style) in Fig.2, relating the couples of runs with the same data.

x) Label sizes in figure 5 are too small

x) The topic of the spurious correlations is brought up for some hand-waving arguments and for justifying some experiments, but those are never actually used for any theory, so I don't see the need of defining the attributes $a_i$.

x) The appendix lacks text describing the figures.

x) The description of Fig.6 in page 4 is confusing. I am not even sure that Figure 6 has the content that is being described.

x) There are many typos in the text, I suggest a careful rereading. Here are some examples:
- forgettabiliy
- Spurious features has been
- neural network experience
- When an exampleS is learned
- and has been used [should be "have"]
- for every exampleS
- we independently generateS
- and dropS many examples
- Subset found by our method allow
- in-distribution. and out-of-distribution

Quality:

The paper is based on an idea that I find nice, but I feel that the paper could have developed and explored it more in depth.


Novelty:

The main message of the paper is a recipe to identify subpopulations within each class, and to exploit this knowledge for better training. As far as I know, this is novel.

Reproducibility:

- The code is not provided, hyperparameter tuning is not explained (nor is it stated how the HPs related to the authors' method should be chosen or how they influence the performance) and the details of the models are not given.
- I don't fully understand how the out of distribution experiments were brought through.



**Strength And Weaknesses:**

Strengths:
- The idea of selecting subpopulations based on the gradients during the initial phase of learning is a very nice effective way of understanding data. This could even be expanded to make static studies of datasets.
- The proposed method seems to outperform previous similar methods

Weaknesses (and further open questions that could/should be addressed in the paper):
1) The argument of section 4.1 leverages on an intuition provided by Nakkiran 2019, the knowledge that minority examples are learned later (which is known, see e.g. arXiv:2104.01769 and arXiv:2207.00391), and the empirical evidence provided in Fig.2. This evidence is provided for a single architecture on a single data set, which is clearly too little to make a general claim.
2) The paper contains mainly empirical results, but the results in section 4 are shown only with a single model.
3) The results in section 5.1 are obtained on one single model and simple datasets. Why not use e.g. the superclasses of cifar-100, with the single classes representing the single populations? In that case there would be no explicitly-imposed spurious correlation, but they can be implicit (i.e. not imposed by hand) and in any case I don't fully understand the need of imposing clear sources of spurious correlations to validate the algorithm (beyond that they allow to show in fig4 that some times spurious correlations are avoided). If the algorithm is meant to help in generic application cases, then I would assume that its efficacy would be visible for any data set, and not only very specific ones, right?
4) There is no analysis of the obtained subpopulation clusters. It is not clear whether the result of the clustering depends on the specific choice of architecture and HyperParameters. Do I obtain the same populations if I change architecture or hyperparameters? Are these found subpopulations actually significant? Analyzing them could confirm whether the arguments leading to the method are correct, or if different explanations should be sought.
5) As I understand from Alg.1 (this was not too clear to me from the main text), the subpopulations are collected in an initial dummy run that lasts t steps. How is t chosen and how do the clusters/performances change with t?
6) The authors select given examples with a probability p_i which depends on the subpopulation [page 6]. What would happen if they didn't use these weights for selection, but rather for reweighting of the single examples?
7) Relatedly, the suggested recipe essentially consists in downsampling the dataset in a smart way, and using a subset S of data instead of the full dataset. One could instead use the probabilities p_i to select the minibatches, and assign examples to a batch depending on p_i. I assume that this would have the same effect as downsampling the dataset, but with the advantage that no example is completely discarded, right?
8) The arguments given by the authors focus on the initial linear model that is created in the first steps of the dynamics. Does this mean that algorithm 1 is not necessary after the beginning of the dynamics?
9) In Fig.3 it is shown that the norms of the examples belonging to the largest clusters are smaller. Going further, shouldn't one be looking for each cluster to have the same total norm (i.e. GradNorm*ClusterSize=constant)?
10) Section 5: why are there no error bars for CMNIST?
11) Section 5.1: how do the clusters found through the authors' method overlap with the real clusters?
12) The following statement is not supported by the experiments and should be restated: "training on the subset selected by our method gives much higher in-distribution test accuracy than Random, and EL2N or forgetting scores particularly when the subset is small". I would rather say that for high pruning -when the subset is small- this is true (though the adjective "high" is arbitrary: I would rather say that this is "consistent" or "systematic" through datasets). For low pruning this is not even true.
Furthermore, the x axes of figures 5a and 5c are skipping the 0.4 point, giving more visual importance to the highly-pruned runs (where the authors' method outperforms the others) than they actually should have. I hope this was not intentionally done to give a false perception of the results.
13) From Figure 5, I see that using the whole dataset is still better than using a subset. Therefore, it would seem that the title ("When majorities prevent learning") is misleading, since keeping the majorities is still good.
14) Beyond the metrics used in table 1, why is the bare accuracy or macro-F1score not reported? As a reader I would be interested in knowing how the authors' method compares to a vanilla training as far as the traditional metrics are concerned. If "majorities prevent learning", do I get an advantage on the global by removing these majorities? Do I lose something? How much? I think these questions should be addressed.




**Summary Of The Paper:**

The authors address the problem that, in a dataset, examples within classes can be grouped into subpopulations that are similar to each other, and this influences the dynamics. They propose an intuitive solution, which does not require previous knowledge on the subpopulations. This solution consists of looking at the gradients related to the single during the initial stages of learning. These gradients are expected to be similar for elements belonging to a single subpopulation, and this allows to identify them through a simple clustering algorithm. The authors test the method on two tasks: worst-group generalization in datasets with spurious correlations, and out-of-distribution generalization under distribution shift. They find promising results in their experiments.



**Summary Of The Review:**

I like the idea underlying the main message of the paper, i.e. separating the examples in subpopulations through the gradients in the first stages of learning, and would be happy to see it published at some point. However, in my view, the current elaboration and presentation do not meet the standards of ICLR. I am open to increase my evaluation if my principal concerns in the Weaknesses and Reproducibility sections are addressed in a convincing manner.


---------------------------------

After rebuttal:

In summary, I  think that the paper has some interesting information, and I appreciate that some of my points were addressed directly, many of which I believe improve the quality of the paper. I changed the evaluation to 6.

---------------------------------

After second round of comments:

All my comments were addressed seriously. I cannot see a new version of the manuscripts, but given the thoroughness of their replies in the forum I have little doubt that the last version will be good. I changed the evaluation to a 7 (I've left a 6 because openreview does not allow for 7).

---

> ### Author Response · Authors · 2022-11-19
> **Response to Reviewer gnhj**
>
> We thank the reviewer for acknowledging the novelty and effectiveness of our algorithm and for the valuable feedback.  Below and in our revised version, we address all the questions and concerns mentioned in the review.
>
> **Evidence for various architectures and datasets:** Our observation and argument hold for different architectures and datasets. We note that different architectures have different training dynamics, which affect the gradient clusters and the initial linear function learned by the model. However, the large gradient clusters *of the corresponding architecture* are learned early and shape the initial linear model learned by *that architecture*. Hence, our argument is valid across architectures. We added new experiments in Appendix Figure 11 showing that the same argument holds when training ResNet-18 on Cifar10. Notably, *larger architectures with more capacity have larger gradient clusters*, which is in line with observations of Sagawa et al that overparameterization exacerbates spurious biases. This is because of their larger capacity to learn more difficult examples and include them in clusters with small gradient norms.
>
> **The choice of the datasets:** The reviewer is right that our method can be applied to general datasets without explicitly-imposed spurious correlation. We chose CMNIST and Waterbirds as they are two benchmark datasets for training with spurious correlations, and are used in all recent publications on the topic (Sagawa et al., 2019; Sohoni et al., 2020; Nam et al., 2020; Ahmed et al., 2020; Liu et al., 2021; Creager et al., 2021; Taghanaki et al., 2021; Zhang et al., 2022; Nam et al., 2021). We used different models to train on these two datasets, i.e. LeNet-5 for CMNIST and ResNet-50 for Waterbirds,  to be consistent with existing work. We agree that discovering spurious correlations is interesting and hence had visualizations of the examples in small and large clusters in CIFAR10 in the Appendix Figure 8. For CIFAR100 superclusters suggested by the reviewer, note that CIFAR100 is a pretty balanced dataset, with only 500 examples in each class. Hence, it does not have a significant bias or spurious correlation, and the accuracy of different classes is very similar. Besides, due to the small size of the classes, there are not many easy examples (which form large gradient clusters) that can be pruned. So, any improvement will be very marginal.
>
> **Analysis of the obtained subpopulation clusters:** As discussed above, the gradient trajectory clusters are architecture-dependent. However, regardless of the choice of the architecture, examples in the large gradient clusters are learned in the initial training epochs and can be identified and pruned by our method. In general, larger models can learn more difficult examples that will be added to larger gradient clusters with a smaller norm. Fig 12 in the Appendix shows that *larger architectures have larger gradient clusters*. We added a new ablation study on the choice of the number of clusters in Appendix Figure 9 (c)-(d), and the choice of the initial epoch we find the clusters at, as we discuss below.
>
> **Choice of the initial epoch:** Our method is not very sensitive to the choice of the initial epoch (as long as it is within the first few epochs). We added a new ablation study in the Appendix Figure 9 (a)-(b) on the choice of the initial epoch. In all our experiments we found the gradient trajectory clusters at epoch 4, without further tuning.

---

> > ### Author Response · Authors · 2022-11-19
> > **Response to Reviewer gnhj (Cont.)**
> >
> > **Reweighting vs upsampling vs downsampling:** We thank the reviewer for the great suggestions. We ran new experiments by (1) upweighting examples in every gradient cluster by the inverse of the cluster size, (2) upsampling the examples from the clusters by the inverse of the cluster size (a simpler version of the suggested mini-batch selection based on $p_i$), and (3) reweighting examples in the clusters such that the clusters have the same gradient total gradient norm. We report the numbers here and have also added the upweighting results to Table 1.
> >
> > We see that upweighting by the inverse of the cluster size works best and achieves a better worst-group and average accuracy than we originally had in Table 1. Upsampling by inverse cluster weight probability performs similarly to ERM, and reweighting by gradient norm does not achieve a competitive performance. This is because upsampling cannot make the clusters entirely balanced, and reweighting based on gradient norms is sensitive to the variance of the gradient norms within the clusters. That is, outlier gradient norms within a cluster can affect its total norm considerably and make reweighting to be inaccurate.
> >
> > |      | CMNIST || Waterbirds ||
> > | ----------- | ----------- | ----------- |----------- | ----------- |
> > |      | Worst-group Acc | Average Acc | Worst-group Acc | Average Acc |
> > | Upsample | $72.4 \pm 1.7$ |$91.4 \pm 1.2$ | $67.7 \pm 2.3$ | $97.3 \pm 0.3$ |
> > | Upweight | $91.6 \pm 1.0$ | $ 96.0 \pm 0.7$ | $79.1 \pm 0.9$ | $97.4 \pm 0.1$ |
> >
> >
> > **Time of applying Alg 1:** The reviewer is right that we only apply Alg 1 once at an initial epoch (we used epoch 4), and train only on the sampled subset afterwards.
> >
> > **Error bars for CMNIST and Figure 5(d):** We added error bars for these two experiments to Table 1 and Figure 5(d) in our revision. We thank the reviewer for pointing it out.
> >
> > **Good in-distribution performance for high pruning:** We thank the reviewer for pointing this out and have revised to: our method achieves a superior in-distribution performance to the baselines when the pruning ratio is high.
> >
> > **When majorities prevent learning:** It is important to distinguish the effect of the “amount of data” vs the “quality of the data”. In Fig 5(c) training on full data is not comparable to smaller subsets, as it involves training on *more examples*. It is, however, fair to compare the performance for subsets of the same size. Random subset has the same distribution as that of the full data. Hence, random subsets of the same size can be compared to our subset. While training on such a subset provides the best in-distribution performance, the bias of the majority subpopulations *prevents learning features of the minority groups as well as core features of the majority subpopulations* (e.g. in case of spurious features), which results in poor worst-group and out-of-distribution performance. This motivates our title.
> >
> > **Bare accuracy or macro-F1 score for Table 1:** Note that for the two datasets used in Table 1, the distribution of training data and test data is very different. Hence, the bare test accuracy does not reflect the quality of the model. Therefore, the adjusted average accuracy is considered in all existing work that try to eliminate the effect of spurious features, as the metric that reflects global performance when the training and testing data comes from the same distribution (Sagawa et al., 2019; Sohoni et al., 2020; Nam et al., 2020; Ahmed et al., 2020; Liu et al., 2021; Creager et al., 2021; Taghanaki et al., 2021; Zhang et al., 2022; Nam et al., 2021). We have reported bare accuracy for standard computer vision benchmarks in Figure 5.
> >
> > **Presentation:** Based on the feedback, we changed the color code in Figure 2, and fixed the description of figures in the Appendix. We also corrected typos.
> >
> > **Reproducibility:** We uploaded the code used for our experiments in our revision.
> >
> > **Out-of-distribution experiments:** We added details of this evaluation to the end of Section 5.2.

---

> > > ### Comment · Reviewer_gnhj · 2022-12-06
> > > **Thanks for your answers**
> > >
> > > I posed my comments as a numbered list and answer according to my numbering.
> > >
> > >
> > > (0) The appendix should be written with more care, with some main text and explanations. The authors reference figures in the appendix and it always takes me a while to understand their content.
> > >
> > > (3) I think I was misunderstood with cifar100. This dataset has 100 classes with 500 images each, which can be grouped into 10 superclasses with 5000 images each. My suggestion was to train on the superclasses, so there is enough data. Now, I would like to know whether and how well the clustering would identify each of the 10 classes in each superclass.
> > >
> > > (3) My doubt, which I do not feel reassured about, is that blatant spurious biases as backgrounds can be detected by the method, but that the method stops there. My reason to think this is that while the gradients at the beginning of learning might capture subpopulations with different backgrounds, many spurious biases are subtle and might not appear represented in the gradient directions. I would like to see how solid the method is, so that the reader can be reassured that it will work for their application case.
> > >
> > > (4) The authors say that the clustering affects any architecture differently, so the clusters created on the same dataset are different when we change model. I guess this is based on the single comparison in the new fig.12 in the appendix, right? While I appreciate the comparison, they are only comparing the number of clusters, but are the clusters completely different, or are the ResNet clusters just a subgrouping of the CNN clusters? I don't understand why not to show this.
> > >
> > > The title still feels like a clickbait, and I still think it should be changed. The argument the authors give on the quality of the data is true, but it does not mean that they "prevent" learning, but rather that more of those examples don't help learning or slow it down (see e.g. arXiv:2104.01769 or arXiv:2207.00391). In order to say that the majority prevents learning I would need plots showing that randomly removing examples from the majority the overall performance (not local indicators) increases.
> > >
> > > In summary, I still think that the paper has some interesting information, and I appreciate that some of my points were addressed directly, many of which I believe improve the quality of the paper. I changed the evaluation to 6 in appreciation of the authors' efforts, though I am unsure about this, because both the presentation and depth of the analysis seem to me borderline with the standards of ICLR.

---

> > > > ### Author Response · Authors · 2022-12-12
> > > > **Thank you and further clarifications**
> > > >
> > > > Thank you very much for taking the time and reading our responses. We appreciate that you considered updating your score and clarify on the new questions below:
> > > >
> > > > (3) We emphasize that our method intends to **identify _only_ the majority subpopulations to _alleviate_ their bias on training**. In fact, the reason that we use early training dynamics is that only large subpopulations are learned _**early**_, and small subpopulations are learned later. We note that there is no need to alleviate the effect of small groups on learning as a small number of similar examples are required to learn the small subpopulations in a generalizable manner, and our method does not aim to identify them.
> > > >
> > > > While our method is not meant to find classes (instead we aim to find majority subpopulations of every class), based on the reviewer's suggestion we ran a new experiment on 20 superclasses of CIFAR100 to find 100 clusters. Our results show that in 82 out of 100 clusters found by our method based on early gradient trajectories, the majority of the examples belong to the same class. The 18 classes that spread in different clusters without being majority in any of them are harder-to-learn classes with many ambiguous examples that require a much longer time to be learned, such as seal/beaver/otter, girl/woman, raccoon/porcupine, seal/beaver, aquarium_fish/flatfish.
> > > >
> > > > Notably, we identified one very large cluster containing 1453 examples, and found that all 1453 examples in this cluster are trees with green leaves, including 304 maple trees, 422 oak trees, 243 palm trees, 251 pine trees, and 233 willow trees (each type of tree is a class in CIFAR100). The images in this large cluster are “green-leaf trees on a white background” in the “trees” supercluster. In the following anonymous GitHub, we uploaded two images in an anonymous GitHub repo (https://anonymous.4open.science/r/iclr23_rebuttal-3075/): the first image (cifar100sup_large.png) shows examples of trees in the largest cluster, and the second image (cifar100sup_small.png) shows examples of trees spread in other clusters. This shows that the majority group in the “trees” superclass can be successfully found by our method.
> > > >
> > > > (3) The reviewer is right that our method does not intend to find subtle spurious correlations. Note that existence of a subtle spurious correlation in the data is not enough for the model to learn it. The model only learns the spurious feature over the core features, if the spurious feature is easier to learn for the model. This only happens when the spurious correlation exists frequently in the data (Sagawa 2020). Our objective is to identify **biases of large subpopulations** in the data, which can be learned by the model and bias its performance.
> > > >
> > > > (4) Note that different models are affected by spurious correlations to different extents, Some are more affected and some are less affected. That is, the same set of examples may bias one model but not the other. (Sagawa et al 2020) showed that overparameterized models are more affected by spurious correlations. This is why spurious correlations need to be addressed for every model differently. Our method finds **majority subpopulations that bias any given model**. Note that, the majority subpopulations, i.e. large clusters of examples with similar gradient trajectories also **depend on the model**. While our algorithms can always find the majorities in the datasets in large gradient clusters, we do not expect or **need** to have the exact same clusters for different architectures because the exact gradient clustering depends on the model architecture therefore, the clustering may not be the same even for each individual run with the same for different network architectures. However, we do find the same majority examples across runs if we compare examples in the _**union of the largest clusters**_. Subsampling examples in large clusters or oversampling examples in small clusters eliminates the bias, without the need for the clusters to be exactly the same for different architectures. Note that our method does not need a threshold to identify large vs. small clusters, and uses the inverse of the cluster size for sampling an unbiased subset.
> > > >
> > > > Below, we show on the Waterbirds dataset that with different models (ConvNet, ResNet-18 and ResNet-50), our algorithm finds the same set of examples in the largest clusters and most of these examples are in the majority groups of Waterbirds.

---

> > > > > ### Author Response · Authors · 2022-12-12
> > > > > **Thank you and further clarifications (Cont.)**
> > > > >
> > > > > In the Waterbirds training set, the majority groups are landbirds on landbackground (LB-L) and waterbirds on waterbackground (WB-W). We have the number of examples in each of the four groups in our paper and we also list them below:
> > > > >
> > > > > | | land background | water background
> > > > > |---|---|---|
> > > > > |landbird | 3498 | 184 |
> > > > > |waterbird| 56 | 1057|
> > > > >
> > > > > From the statistics, we can see that there are 3498+1057=4555 examples in the majority groups. To compare across models and also with the ground truth groups, we select examples in clusters sorted from largest to smallest, as long as we have approximately 4555 examples. As we can see from the tables below, CNN and ResNet-18 have 4194/4200=99.9% examples overlapped in their large clusters, and 4141/4194=98.7% of these overlapping examples are from the ground-truth majority groups of Waterbirds. Similarly for ResNet-18 and ResNet-50, 4055/4203=96.5% examples overlapped in their large clusters, and 4018/4055=99.1% of these overlapping examples are from the ground-truth large groups of Waterbirds. These results show that we find the same examples in the underlying majority groups in the dataset with different models.
> > > > >
> > > > > | | CNN | ResNet-18 | Overlap |
> > > > > |---|---|---|---|
> > > > > |Large clusters| 4200 | 4203 | 4194 (4141 from LB-L or WB-W) |
> > > > > |Small clusters | 595 | 592 | 586 |
> > > > >
> > > > > | | ResNet-50 | ResNet-18 | Overlap |
> > > > > |---|---|---|---|
> > > > > | Large clusters | 4221 | 4203 | 4055 (4018 from LB-L or WB-W) |
> > > > > | Small clusters | 574 | 592 |426 |
> > > > >
> > > > > We respectfully disagree with the reviewer’s statement about the examples in the majority groups, that “more of those examples don't help learning or slow it down”. Examples in majority groups bias the model towards themselves and this bias remains in the model until convergence (Nakkiran et al 2019). This results in overfitting (i.e., **memorizing**) the examples in the minority groups instead of learning them in a _**generalizable**_ manner, even after training for a long time (Sagawa et al 2020). While we cannot update the manuscript at this point, we will change our title in the revision in a way that better reflects our contributions.

---

> > > > > > ### Comment · Reviewer_gnhj · 2022-12-12
> > > > > > **I am ok with publication**
> > > > > >
> > > > > > I appreciate the authors taking my comments seriously into account.
> > > > > >
> > > > > > (3) The results on cifar100 reflect what one could expect: in many cases, the clustering sees the individual classes, while in other cases (the different kinds of tree classes being coarsened into a single category), according to the classifier at the beginning of the dynamics, the classes should be grouped in a different way than they are in the dataset. I think it would be interesting to have this described in the final version of the manuscript.
> > > > > >
> > > > > > (3) Given the example here above, I understand better the autors' point. It would be about having the trees-with-white-background pseudoclass stand out less during the learning. Do the authors think that if they applied their method to this cifar100 example, they would obtain an increased performance?
> > > > > >
> > > > > > (4) I see the authors' point. Although different models might have different spurious correlations, intuition would be that those driving the dynamics at the beginning of learning would be similar and independent of the model. This seems to be confirmed by the extra runs on the waterbirds dataset.
> > > > > >
> > > > > > I also appreciate the authors reconsidering the title.
> > > > > >
> > > > > > I thank the authors for the extra analyses. I would add them, as well as the resulting considerations, to the new version of the manuscript.

---

> > > > > > > ### Author Response · Authors · 2022-12-13
> > > > > > > **Thank you!**
> > > > > > >
> > > > > > > Thank you for following our response and providing additional feedback. We really appreciate it.
> > > > > > >
> > > > > > > (3) The CIFAR100 experiment was indeed a nice example and a great suggestion. We thank the reviewer and will definitely add this example to the revised version of the manuscript.
> > > > > > >
> > > > > > > (3) Yes, we believe balancing the clusters in the “trees” superclass will improve the performance of the minority tree clusters. We did not have enough time to finish the experiment by today, but will add the results to our revised version.
> > > > > > >
> > > > > > > (4) That’s correct. We will clarify this better in the revision.
> > > > > > >
> > > > > > > We will include all the discussion in the new version of the manuscript. Thank you again for all your valuable comments provided in this discussion period. They helped to improve the quality of our manuscript a lot.

---

### Official Review · Reviewer_d2Nd · 2022-10-24

**Confidence:** 4
**Correctness:** 3
**Technical Novelty And Significance:** 3
**Empirical Novelty And Significance:** 3
**Recommendation:** 5

**Clarity, Quality, Novelty And Reproducibility:**

Clarity: The writing is easy to understand, but there are many typos.
Novelty: It seems quite novel to me using gradients for identifying biased groups.
Reproducibility: Code and hyperparameters are not provided.

**Strength And Weaknesses:**

Strength:
1. An elegant solution for an important problem. DRO is intractable in deep learning and manually crafted groups are introduced to solve this problem. Automatic 1) selection of shared-feature groups, 2) identification of data with spurious features, and 3) debiasing has been an unsolved problem. This paper uses gradient-related statistics to identify majority groups and can inspire many future works in this area.

2. The experiment results are competitive, especially for debiasing and OOD generalization.

Weakness:
1. The motivation has not been stated clearly enough. The authors in fact introduce three concepts "long-tail"->"spurious features & debiasing"->"OOD & worst-case generalization".  The link between majority group (long tail) and improved generalization is a bit weak and the author could better elucidate this connection. You may find these papers [1] [2] helpful on this subject.

2. I'm confused by the motivation for using gradient trajectories to identify majority groups. "In the initial training epochs, the network learns important features and the NTK undergoes rapid changes, which determine its final basin of convergence" and "As a result, the large subpopulations dictate the rapid initial change of the NTK and the prominent features learned in this phase". The authors seem to be motivated by the dynamics of NTK for identifying majority groups. Therefore, I'm confused why they chose gradient trajectory rather than gram matrix trajectory as the indicator.

3. Problem with Lemma 4.1
- I think the result is not specific to this particular choice of grouping based on gradient. Actually, it seems to be only affected by group size and this result is rather trivial as balanced groups would be less affected by reweighting.

4. Experiments
- ERM is a fairly weak baseline, and comparison with DRO and EL2N are missing in Table 1.
- Gradient trajectories are instable and sensitive to hyperparameters. Sensitivity analysis and error bars are missing (CMNIST, Figure 5d). Also, a large variety of models should be chosen and analyzed, as different backbones have very different training dynamics.
- Code and hyperparameters are not provided.

[1] Ming Y, Yin H, Li Y. On the impact of spurious correlation for out-of-distribution detection[C]//Proceedings of the AAAI Conference on Artificial Intelligence. 2022, 36(9): 10051-10059.

[2] Tang, Kaihua, et al. "Invariant feature learning for generalized long-tailed classification." arXiv preprint arXiv:2207.09504 (2022).

**Summary Of The Paper:**

This paper proposes a method to up weight the smaller sub-populations that are found by performing clustering of gradient trajectories. This method is improved upon Group-DRO, which automatically identifies groups and prevents overfitting on spurious correlations. In order to achieve that, they use the premise that spurious features are abundant in majority groups and large subpopulations with similar gradient trajectories can be identified based on their gradient trajectory during the first few epochs. Their main contribution is to introduce an automatic clustering process that identifies large groups with spurious features and eliminating these biases improves worst-case and OOD performance. The authors validates the effectiveness of their methods on crafted datasets (Waterbird and CMNIST) and general image classification datasets ( CIFAR and Caltech-256). Improvements are shown in both classification accuracy and Gradcam saliency.



**Summary Of The Review:**

Overall, this paper could be a significant algorithmic contribution, with a few caveats for some clarifications on the theory and experiments. This algorithm can be applied to worst-case robustness, OOD generalization, and can be extended in general to the debiasing scenario. If these problems are sufficiently addressed in the rebuttal, I would lean to accept this paper.

---

> ### Author Response · Authors · 2022-11-19
> **Response to Reviewer d2Nd**
>
> We thank reviewer d2Nd for acknowledging our algorithm as an elegant and effective solution for an important problem and for the valuable feedback.
>
> **Motivation:** We thank the reviewer for pointing us to the two valuable references. Indeed, modeling the data shifts by taking into account both the invariant and environmental (spurious) features, allows unifying our objective, and makes the connection on how spurious correlation in the training set impacts OOD performance. Balancing the subpopulations within a class, formalized as the generalized long-tail problem, nicely captures the objective of our method and explains its effectiveness in improving the OOD performance. We thank the reviewer again and will clarify the connections in our motivation.
>
> **Using gradient trajectories instead of Gram matrix trajectories:** The rapid change of NTK *during the initial training epochs* motivates the importance of debiasing (pruning large subpopulations) *early in training*. During the first few epochs, the initial linear model is formed, and will be preserved until convergence (Nakkiran et al 2019). Hence, if debiasing does not take place during the initial epochs, the bias of the majorities (including the learned spurious features) will be learned and will remain in the model afterwards. Based on this observation, we prune the large subpopulations during the *first few training epochs*.
> In practice, computing the kernel and kernel gram matrix is computationally prohibitive even for medium-sized datasets and networks. Besides, the very large dimensionality of the Jacobian prevents finding examples with similar Jacobian trajectories, as in high dimensional spaces all the points are far away from each other (Aggarwal et al 2001). However, the change in the kernel or gram matrix of the kernel is dictated by the alignment of the Jacobian with the label vector, i.e., the gradient. This motivates the use of gradient trajectories to find majority subpopulations.
>
> **Lemma 4.1:** The RHS of Lemma 4.1 depends on both group sizes and the *maximum gradient norm within every group*. As we demonstrated in Fig 3 and discussed after the Lemma, *examples in large gradient trajectory clusters have consistently smaller gradient norms* during the first few epochs of training. This allows pruning more examples in the large clusters without affecting the model too drastically. This argument does not hold for other choices of grouping, such as random grouping.
> Indeed, when the gradient trajectory clusters are balanced, our pruning only reduces the learning rate. However, when clusters are imbalanced, our method effectively prunes the majority groups and alleviates their bias.
>
>
>
> **Baselines in Table 1:** While El2N works well for preserving the in-distribution performance, it fails to provide a good worst-group performance and is not considered a strong baseline in recent works on alleviating spurious correlations (Sagawa et al., 2019; Sohoni et al., 2020; Nam et al., 2020; Ahmed et al., 2020; Liu et al., 2021; Creager et al., 2021; Taghanaki et al., 2021; Zhang et al., 2022; Nam et al., 2021). We added two baselines for robust training against spurious correlations, namely group DRO (Sagawa et al., 2019) and JTT (Liu et al., 2021), to Table 1. JTT trains the model once on the original data to identify misclassified examples and another time while upsampling the identified examples by 100x. Hence, it is very expensive and sensitive to the choice of hyperparameters (the epoch when the misclassified examples are found, and the number of times they are upsampled). In contrast, our method is very efficient and is not very sensitive to hyperparameters.
>
> For CnC, we couldn’t replicate the numbers reported for Waterbirds in their paper with their official code, pretrained model, and instructions, so we used the reported numbers in this table. We didn’t have problems when running the CMNIST dataset. We will update once we confirm the numbers.
>
> |      | CMNIST || Waterbirds ||
> | ----------- | ----------- | ----------- |----------- | ----------- |
> |      | Worst-group Acc | Average Acc | Worst-group Acc | Average Acc |
> | Group DRO | $66.0 \pm 3.1$ | $92.0 \pm 0.3$ | $83.1 \pm 0.6$ | $92.8 \pm 0.1$ |
> | JTT | $61.6 \pm 5.6$ | $97.2 \pm 0.2$ | $86.0 \pm 1.4$ | $91.5 \pm 0.5$ |
> | CnC | $84.2 \pm 2.4$ | $92.0 \pm 0.4$ | $88.5 \pm 0.3$ | $90.9 \pm 0.1$ |

---

> > ### Author Response · Authors · 2022-11-19
> > **Response to Reviewer d2Nd (Cont.)**
> >
> > **Missing error bars for CMNIST and Figure 5(d):** We added error bars for these two experiments to Table 1 and Figure 5(d) in our revision. We note that gradient trajectories are not insatiable or sensitive to hyperparameters. It is correct that different architectures have different training dynamics, which affect the gradient clusters. However, the large gradient trajectory clusters *of the corresponding architecture* are learned early and shape the initial linear model learned by *that architecture*. Such clusters can be successfully found by our method for various architectures.
> >
> > We added new experiments in Appendix Figure 11 showing that the same argument holds when training ResNet-18 on Cifar10. Notably, *larger architectures with more capacity have larger gradient clusters*, which is in line with observations of Sagawa et al that overparameterization exacerbates spurious biases. This is because of their larger capacity to learn more difficult examples and include them in clusters with small gradient norms.
> >
> > **Reproducibility:** We uploaded the code used for our experiments in our revision.

---

> ### Author Response · Authors · 2022-12-07
> **Looking forward to the reviewer response**
>
> We would like to follow up to see if our response addressed your concerns and if there is anything else we can clarify before the deadline.

---

### Official Review · Reviewer_nnQ5 · 2022-11-05

**Confidence:** 3
**Correctness:** 3
**Technical Novelty And Significance:** 4
**Empirical Novelty And Significance:** 4
**Recommendation:** 5

**Clarity, Quality, Novelty And Reproducibility:**

The Clarity, Quality and Novelty of this paper could be improved. I consider the Reproducibility to be fair as no code has been provided yet.

**Strength And Weaknesses:**

Strength: overall very well written paper with strong and clear motivation.

Weakness: my concern is primarily on the technical side.

Tractable toy examples: While the high-level idea of tracking gradient seems to make sense, the introduction of clustering and dimensionality reduction inevitably introduce a lot of approximation errors into the process. The authors conducted experiments mainly w.r.t CIFAR cases, but it would that make more sense to design some analytical examples to demonstrate the effectiveness of this work, where things stay low-dimensional to begin with.

Importance sampling: the choice of inverse weight seems a bit arbitrary. Have the authors considered an ablation study on the different choices of IS proposals?

Improving the implementations: I don't think ERM is the SOTA baseline to compare with, as it is known to have sub-par performance. In addition, the authors might want to conduct more ablation studies w.r.t. the gradient tracking, as some of the techniques adopted are quite sensitive to seed initialization (the k-means clustering, for example).

**Summary Of The Paper:**

Good generalization performance of DL could be attributed to the undesirable overfitting spurious biases in large datasets. This apper proposed a way to conduct smart sub-sampling by tracing the gradient trajectories of initial examples. An importance sampling algorithm is then proposed to conduct sub-sampling.

**Summary Of The Review:**

I would like to see the rebuttal of this paper and reconsider my standpoint, but currently, I incline to reject as is, unless a better explanation and validation of the gradient tracking technique is added.

---

> ### Author Response · Authors · 2022-11-19
> **Response to Reviewer nnQ5**
>
> We thank reviewer nnQ5 for the valuable feedback.
>
> **Analytical Example:** Consider a 2-dimensional linear regression $y_i=<w,x_i>$, with 3 groups of examples: there are (1) a group of $a_1$ examples with features $x_i=(1,0)$ and $y_i=0$; (2) a group of $a_2$ examples with $x_i=(-1,0)$ and $y_i=0$, and (3) a group of $a_3$ examples with $x_i=(-1,1)$ and $y_i=1$. For $w_0=0$, the gradient of examples in the first and second group is $0$, and the gradient of examples in the third group is $1$. Gradient descent updates the model based on the average of the gradient of all examples, i.e, $a_3/(a_1+a_2+a_3)$. If $a_1, a_2 >> a_3$, the average gradient becomes nearly 0 and the model does not learn the third group. However, if $a_1 \sim a_2 \sim a_3$, the model will learn the second and third group equally.
> For several classes of convex problems, including linear regression, ridge regression, logistic regression, and regularized support vector machines (SVMs), the normed gradient difference between data points can be efficiently boundedly approximated by the normed difference between their feature vectors (Allen-Zhu et al., 2016; Hofmann et al., 2015). Hence, examples with similar features have similar gradients, and groups of examples with similar features will be recovered by our clustering algorithms, and become relatively biased with our importance sampling method.
>
> Sampling by the inverse weight probability: As discussed in Sec 4.3, sampling by inverse weight probability is intended to select a sample with an equal number of examples from different clusters. In doing so, our method alleviates the bias in the data. We added new experiments with square root of the inverse weight, and upweighting the examples, which we report here.
> |      | CMNIST || Waterbirds ||
> | ----------- | ----------- | ----------- |----------- | ----------- |
> |      | Worst-group Acc | Average Acc | Worst-group Acc | Average Acc |
> | Downsample by Sqrt Inverse Size | $72.2 \pm 2.7$ |$92.8 \pm 0.4$ | $73.4 \pm 1.1$ | $97.2 \pm 0.1$ |
> | Upweight | $91.6 \pm 1.0$ | $96.0 \pm 0.7$ | $79.1 \pm 0.9$ | $97.4 \pm 0.1$ |
>
> **Ablation study on gradient tracking:** Please note that we use k-medoids and not k-means to find the gradient trajectory clusters. Importantly, we use the submodular formulation for the k-medoids problem, and find the medoids (most centrally located gradient trajectories) using a (deterministic) greedy algorithm (Alg 1, lines 4-6 shows the greedy selection of the medoids, and lines 7-8 shows finding the clusters). Hence, unlike k-means that, as noted by the reviewer, is sensitive to the choice of initial seeds, our k-medoids clustering does not involve any randomness or hyperparameter. That is, for a fixed set of gradient trajectories, we always get the same fixed clusters.
>
> **Additional baselines other than ERM:** We added three recent baselines for robust training against spurious correlations, namely group DRO (Sagawa et al., 2019), JTT (Liu et al., 2021) and CnC ( Zhang et al., 2022). Note that JTT is very expensive as it requires training the model once on the original data and another time on the upsampled data that is orders of magnitude larger. It is also sensitive to the choice of hyperparameters (the epoch when the misclassified examples are found, and the number of times they are upsampled). Hence, JTT is not scalable to large datasets. In contrast, our method is very efficient and does not require any hyperparameter tuning.
> For CnC, we couldn’t replicate the numbers reported for Waterbirds in their paper with their official code, pretrained model, and instructions, so we used the reported numbers in this table. We didn’t have problems when running the CMNIST dataset. We will update once we confirm the numbers.
>
> |      | CMNIST || Waterbirds ||
> | ----------- | ----------- | ----------- |----------- | ----------- |
> |      | Worst-group Acc | Average Acc | Worst-group Acc | Average Acc |
> | Group DRO | $66.0 \pm 3.1$ | $92.0 \pm 0.3$ | $83.1 \pm 0.6$ | $92.8 \pm 0.1$ |
> | JTT | $61.6 \pm 5.6$ | $97.2 \pm 0.2$ | $86.0 \pm 1.4$ | $91.5 \pm 0.5$ |
> | CnC | $84.2 \pm 2.4$ | $92.0 \pm 0.4$ | $88.5 \pm 0.3$ | $90.9 \pm 0.1$ |
>
> **Reproducibility:** We uploaded the code used for our experiments in our revision.

---

> ### Author Response · Authors · 2022-12-07
> **Looking forward to the reviewer response**
>
> We would like to follow up to see if our response addressed your concerns and if there is anything else we can clarify before the deadline.

---

### Decision · Program_Chairs · 2023-01-20

**Decision:**

Reject

**Justification For Why Not Higher Score:**

Three out of four reviewers raised many concerns that need to be addressed and passed through another round of reviews.

**Justification For Why Not Lower Score:**

N/A

**Metareview: Summary, Strengths And Weaknesses:**

This paper aims to improve worst-group performance and out-of-distribution generalization, and builds upon the group-DRO framework. The conjecture is that large (majority) subpopulations can be identified based on gradient trajectory during the first few epochs. The paper performs clustering of similar gradient trajectories to identify these large subgroups and subsample them to rebalance the dataset, and to improve worst-case and OOD performance.

While acknowledging that the proposed model is potentially useful, the reviewers have provided detailed comments and raised many concerns -- see weakness sections of all reviewers, especially Reviewer gnhj, Reviewer BcrE. Most critical are related to 1) empirical evaluations that are required to assess the scope and significance of this work contributions, including strong baselines, and 2) writing and presentation clarity of the paper. Extensive rebuttal convinced one of four reviewers to vote for acceptance. The reviewer acknowledged, however, that they haven’t seen a revised version of the manuscript. The manuscript will further benefit from an extended related work that highlights similarity and differences when benchmarking ODD approaches to clarify any confusions.
AC decides that albeit it’s an important study, the manuscript will require too many revisions to be prepared and verified at this stage, and will benefit from another round of reviews. We hope the reviews are useful to improve the manuscript.